# Gut Microbiome-Targeted Modulations Regulate Metabolic Profiles and Alleviate Altitude-Related Cardiac Hypertrophy in Rats

Yichen Hu,[a] Zhiyuan Pan,[b] Zongyu Huang,[b] Yan Li,[a] Ni Han,[b] Xiaomei Zhuang,[c] Hui Peng,[d] Quansheng Gao,[d] Qing Wang,[e] B. J. Yang Lee,[f] Heping Zhang,[g] Ruifu Yang,[b] Yujing Bi,[b] Zhenjiang Zech Xu[a,h]

[a]State Key Laboratory of Food Science and Technology, Nanchang University, Nanchang, People's Republic of China
[b]State Key Laboratory of Pathogen and Biosecurity, Beijing Institute of Microbiology and Epidemiology, Beijing, People's Republic of China
[c]Institute of Pharmacology and Toxicology, Beijing, People's Republic of China
[d]Tianjin Institute of Environmental and Operational Medicine, Tianjin, People's Republic of China
[e]Beijing Hi-LongCare Medicine & Technology Co., Ltd., Beijing, People's Republic of China
[f]Beijing Future Science & Technology Development Co., Ltd., Beijing, People's Republic of China
[g]Key Laboratory of Dairy Biotechnology and Engineering, Education Ministry of People's Republic of China, Department of Food Science and Engineering, Inner Mongolia Agricultural University, Hohhot, People's Republic of China
[h]Microbiome Medicine Center, Department of Laboratory Medicine, Zhujiang Hospital, Southern Medical University, Guangzhou, Guangdong, China

Yichen Hu, Zhiyuan Pan, and Zongyu Huang contributed equally to this article. Author order was determined by the corresponding author after negotiation. Heping Zhang and Ruifu Yang are senior authors.

**ABSTRACT** It is well known that humans physiologically or pathologically respond to high altitude, with these responses accompanied by alterations in the gut microbiome. To investigate whether gut microbiota modulation can alleviate high-altitude-related diseases, we administered probiotics, prebiotics, and synbiotics in rat model with altitude-related cardiac impairment after hypobaric hypoxia challenge and observed that all three treatments alleviated cardiac hypertrophy as measured by heart weight-to-body weight ratio and gene expression levels of biomarkers in heart tissue. The disruption of gut microbiota induced by hypobaric hypoxia was also ameliorated, especially for microbes of *Ruminococcaceae* and *Lachnospiraceae* families. Metabolome revealed that hypobaric hypoxia significantly altered the plasma short-chain fatty acids (SCFAs), bile acids (BAs), amino acids, neurotransmitters, and free fatty acids, but not the overall fecal SCFAs and BAs. The treatments were able to restore homeostasis of plasma amino acids and neurotransmitters to a certain degree, but not for the other measured metabolites. This study paves the way to further investigate the underlying mechanisms of gut microbiome in high-altitude related diseases and opens opportunity to target gut microbiome for therapeutic purpose.

**IMPORTANCE** Evidence suggests that gut microbiome changes upon hypobaric hypoxia exposure; however, it remains elusive whether this microbiome change is a merely derivational reflection of host physiological alteration, or it synergizes to exacerbate high-altitude diseases. We intervened gut microbiome in the rat model of prolonged hypobaric hypoxia challenge and found that the intervention could alleviate the symptoms of pathological cardiac hypertrophy, gut microbial dysbiosis, and metabolic disruptions of certain metabolites in gut and plasma induced by hypobaric hypoxia. Our study suggests that gut microbiome may be a causative factor for high-altitude-related pathogenesis and a target for therapeutic intervention.

**KEYWORDS** cardiac hypertrophy, hypobaric hypoxia, microbiota, 16S rRNA, metabolome

Address correspondence to Yujing Bi, byj7801@sina.com, or Zhenjiang Zech Xu, zhenjiang.xu@gmail.com.

The authors declare no conflict of interest.

Chronic mountain sickness (CMS) is a syndrome generally observed in populations living in high-altitude regions, which develops when the capacity for hypoxia adaptation is lost (1). As a hallmark feature of CMS (2), hypoxia-induced myocardial hypertrophy is also commonly viewed as the fundamental pathological process of chronic heart failure (3), a leading cause for cardiovascular mortality worldwide (4). During the past decades, several biological theories have been proposed to explain the altitude-related pathological cardiac hypertrophy after long-term hypobaric hypoxia exposure, including oxidative stress (5), inflammation (6), and kinase activity (7). However, the underlying mechanisms are still not completely elucidated (8).

Intriguingly, a growing body of evidence emphasizes the importance of the gut microbiota in human health and diseases (9). Clinical investigations unveil an altered gut microbiome in patients with chronic heart failure (10), characterized by an overgrowth of pathogenic bacteria (11), as well as a decreased abundance of potential health-promoting microorganisms such as species belonging to *Faecalibacterium*, *Lachnospiraceae* (12, 13), *Erysipelotrichaceae* (14) and *Ruminococcaceae* (15), which are known as the producers of SCFAs (16). In a cardiac hypertrophy model induced by post infarction, antibiotics-treated loss of gut microbiota aggravated the severity of myocardial hypertrophy (17, 18); SCFAs (propionate) and probiotics supplementation attenuated the symptoms (17, 19). Dysbiosis of the gut microbiota was also associated with the cardiac damage in hypertensive rat models (20); administration of propionate (21) and probiotics (22) exhibited beneficial anti-inflammatory and anti-hypertensive effects. Likewise, multiple studies suggest that microbiota-derived metabolites (e.g., trimethylamine-N-oxide and bile acids) also involve in the development of chronic heart failure (10, 23, 24). Thus, the gut microbiota might play a critical role in the pathogenesis and progression of cardiac hypertrophy and heart failure (25, 26). However, knowledge is still lacking about the link between gut microbiota and CMS, particularly in altitude-related cardiac hypertrophy.

In a parallel study, we found that prolonged hypobaric hypoxia induced a pathological cardiac hypertrophy in rats, which was associated with significant alternations in gut microbiota and its metabolic potentials. Whether gut microbiome-targeted interventions could ameliorate the development of altitude-related cardiac hypertrophy is therefore warranted.

In this study, we used a factorial design to investigate the effect of probiotics, prebiotics, and synbiotics treatments on hypobaric hypoxia-induced cardiac hypertrophy. We profiled the fecal microbiota, fecal SCFAs and BAs as well as plasma metabolome and cytokines. Our results showed that administration of probiotics, prebiotics, and synbiotics significantly attenuated the cardiac hypertrophy, dysbiosis of gut microbiota and plasma metabolic profiles in hypobaric hypoxia-challenged rats, suggesting that gut microbiota-targeted treatments might be promising therapeutic strategies for altitude-related cardiac hypertrophy.

## RESULTS

**Study design.** As shown in Fig. S1A, the rats were randomly placed in two environments: normobaric normoxia (NN) and hypobaric hypoxia (HH). NN rats were maintained at sea level, while HH rats were kept in hypobaric hypoxia chamber simulating 5,000 m high-attitude environment for 28 days. Our parallel study has proved that prolonged exposure of HH leads to pathological myocardial hypertrophy with profound changes in gut microbiota and metabolites. To determine whether oral administration of probiotics and/or prebiotics could restore the gut microbial ecosystem and attenuate HH-induced cardiac hypertrophy, the rats in both environments were daily gavaged with saline as control, probiotics, prebiotics or synbiotics for 28 days. The probiotics supplement contains 3 *Bifidobacterium* strains (*B. animalis subsp. lactis V9*, *B. longum KT-L9*, *B. adolescentis KT-A8*) and 6 *Lactobacillus* strains (*L. casei Zhang*, *L. plantarum P-8*, *L. paracasei KT-P6*, *L. rhamnosus M9*, *L. acidophilus KT-A1*, *L. helveticus H9*); the prebiotics consists of polydextrose, galactose, and inulin; and the synbiotics is a mixture of the two. We sampled rat stools before (day 0) and after treatments (day 28). Plasma and heart were collected at the end of the experiment.

To verify the presence of probiotics after 28 days of gavage, we utilized specific primers and performed qPCR experiment to detect *L. casei Zhang* and *L. plantarum P-8* strains, as

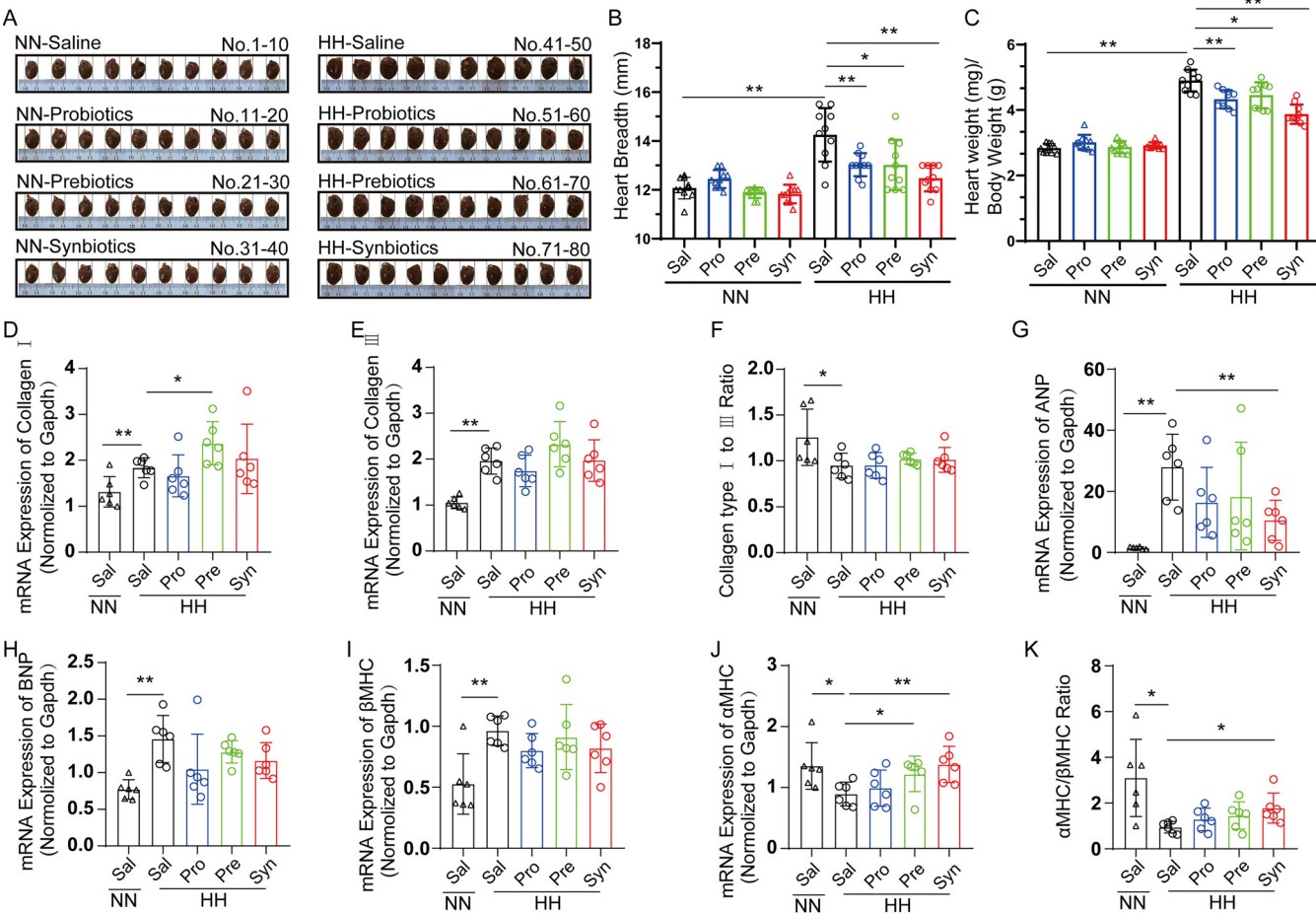

**FIG 1** Gut microbiome-targeted treatments alleviated HH-induced pathological cardiac hypertrophy. (A) Image of rat hearts captured at the end of the experiment. (B, C) Heart breadth (B) and the ratio of heart weight/body weight (C) were measured in each group. (D–K) The mRNA gene expressions of type I collagen (D), type III collagen (E), the ratio of type I collagen/type III collagen (F), atrial natriuretic peptide (ANP) (G), B-type natriuretic peptide (BNP) (H), cardiac myosin heavy chain beta isoform ($\beta$MHC) (I), cardiac myosin heavy chain alpha isoform ($\alpha$MHC) (J), the ratio of $\alpha$MHC/$\beta$MHC (K) at the apex of the heart tissue were detected by quantitative real-time PCR. Data were normalized to the expression of the glyceraldehyde-3-phosphate dehydrogenase (Gapdh) gene. Data are presented as means ± S.D., $n = 6$ rats/group. The statistical significance was performed by GraphPad Prism 8.0 software using one-way analysis of variance followed by Student's $t$ test between groups. * $P < 0.05$, ** $P < 0.01$.

well as *L. rhamnosus*, *L. helveticus*, *B. adolescentis species*, *B. animalis subsp. lactis*, and *Bifidobacterium* genus. Significant higher levels of *L. casei Zhang* (Fig. S2B), *L. plantarum P-8* (Fig. S2C), *L. rhamnosus* (Fig. S2E), *B. animals sub. lactis* (Fig. S2F), and *Bifidobacterium* (Fig. S2H) were observed in HH-probiotics rats on day 28 than that on day 0. We also compared the abundance of *L. casei Zhang* (Fig. S2I), *B. animals sub. lactis* (Fig. S2J) and *Bifidobacterium* (Fig. S2K) on day 28 in both saline and probiotics groups. Their abundances were increased in probiotics group compared to that in saline group under both NN and HH conditions. The results together confirm the presence of probiotics in feces.

**Gut microbiome-targeted treatments alleviated HH-induced pathological cardiac hypertrophy.** Studies have shown that hypoxia could induce cardiac hypertrophy, an adaptive response of heart to increase workload to maintain cardiac homeostasis, which is usually characterized by elevated heart size (27, 28), cardiomyocyte size, and thick ventricular walls (29). A significant increase in heart size (Fig. 1A and B) and heart index (the ratio of wet heart weight to total body weight, Fig. 1C) were observed in HH rats. Interestingly, probiotic, prebiotic, and synbiotic treatments all statistically significantly diminished the increase of heart size (Fig. 1A and B) and heart index (Fig. 1C) induced by hypobaric hypoxia, indicating that they effectively alleviated altitude-related cardiac hypertrophy, while having little influence on rats under NN condition.

Physiological or pathological are two types of cardiac hypertrophy. Differing from physiological hypertrophy with normal contractile function and cardiac structure, pathological cardiac hypertrophy is characterized by myocardial remodeling and finally progress to heart failure. As biomarkers of cardiac fibrosis, type I and type III collagen gene expressions are increased, serving as an important characteristic of progressive heart failure (30, 31). Lower ratio of type I and type III collagen is commonly observed in ischemic cardiomyopathy (32). In our study, we observed higher levels of type I and type III collagen and reduced type I/type III collagen ratio in HH saline group compared to NN saline group, indicating cardiac remodeling and ischemic cardiomyopathy. However, treatments with probiotics, prebiotics, or synbiotics failed to affect these changes in type I (Fig. 1D), type III collagen (Fig. 1E), or their ratio (Fig. 1F). Atrial natriuretic peptide (ANP) and brain natriuretic peptide (BNP) are hormones secreted by the heart and they increase in accordance with the severity of pathological hypertrophy (3). Their mRNA levels were dramatically higher at the apex of the heart tissue in HH saline rats compared to those in NN saline rats. The expression of ANP was slightly downregulated after treatment with probiotics, prebiotics or synbiotics, even though only synbiotics group significantly lessened the ANP level (Fig. 1G). Similarly, the BNP gene expression was also partially restored in the HH treated rats (Fig. 1H). Cardiac myosin heavy chain (MHC) is another hypertrophic marker and is expressed in alpha and beta isoforms (33). $\beta$MHC is often found to be upregulated and the gene expression ratio of $\alpha$MHC/$\beta$MHC isoforms lowered in pathological heart hypertrophy (3). Our results showed that the hypobaric hypoxia challenge decreased $\alpha$MHC (Fig. 1J) and increased $\beta$MHC (Fig. 1I), resulting in a reduced $\alpha$MHC/$\beta$MHC ratio (Fig. 1K). This decreased ratio was moderately reversed after treatments, especially the synbiotics group ($P$ $value$ < 0.05, Fig. 1K), mainly via the upregulation of $\alpha$MHC expression (Fig. 1J). Compared to the HH saline rats, the $\beta$MHC expression levels in the treated rats were not significantly changed, although they appeared a bit lower in the probiotics and synbiotics groups (Fig. 1I). Collectively, our data suggest that gut microbiome-targeted treatments alleviated pathological hypertrophic changes caused by the prolonged HH exposure.

**The treatments ameliorated the effect of hypobaric hypoxia on gut microbiota.** To investigate the gut microbiota changes induced by probiotics, prebiotics, and synbiotics in hypobaric hypoxia-challenged rats, we performed 16S rRNA gene amplicon sequencing of the V4 region on stool samples collected on day 0 and 28. The taxonomy composition consisted mostly of *Ruminococcaceae* followed by *S24_7*, *Lactobacillaceae*, *Lachnospiraceae*, *Verrucomicrobiaceae*, *Prevotellaceae*, and unresolved families from Closteridiales order, which account for more than 80% of the total abundance (Fig. S2), agreeing with previous reports (34). It seems the abundances of *S24_7*, *Lactobacillaceae*, and *Prevotellaceae* were changed by prolonged hypobaric hypoxia exposure. However, the taxonomic difference is not discernible among treatment groups on the family-level taxonomy plot.

We didn't observe significant alpha diversity differences between treatment groups measured with Shannon index, Faith PD, or Observed ASVs (amplicon sequence variants). In line with the literature, age is the predominant factor driving gut microbial composition (35), as shown that samples from the two time points are clearly separated on PCoA plot with either weighted or unweighted UniFrac distance (Fig. 2A). Mantel test also showed that the gut microbiota profiles were not significantly similar between day 0 and day 28 of the same rats, confirming the large divergence in gut microbiota arising from age (unweighted UniFrac, p-val = 0.17; weighed UniFrac, p-val = 0.95). Hypobaric hypoxia challenge also considerably affected gut microbiota, as NN and HH samples of day 28 were clustered into their own groups on the PCoA plot, regardless of their treatments (Fig. 2B). In order to control the age effects and cancel out the baseline variation of individual rats, we computed the log-ratio of day-28 abundance over day-0 abundance for each microbe, and then used the log-ratio values to represent the microbial profiles to avoid the effects of treatments over drowned by the age factor. Our PERMANOVA results showed that the gut microbiota was significantly different between saline group and any other treatment group under either normal or hypobaric hypoxia conditions (Table S1). Furthermore, the divergence in gut microbiota caused by hypobaric hypoxia was significantly alleviated, as suggested by the result that beta

Microbiology Spectrum

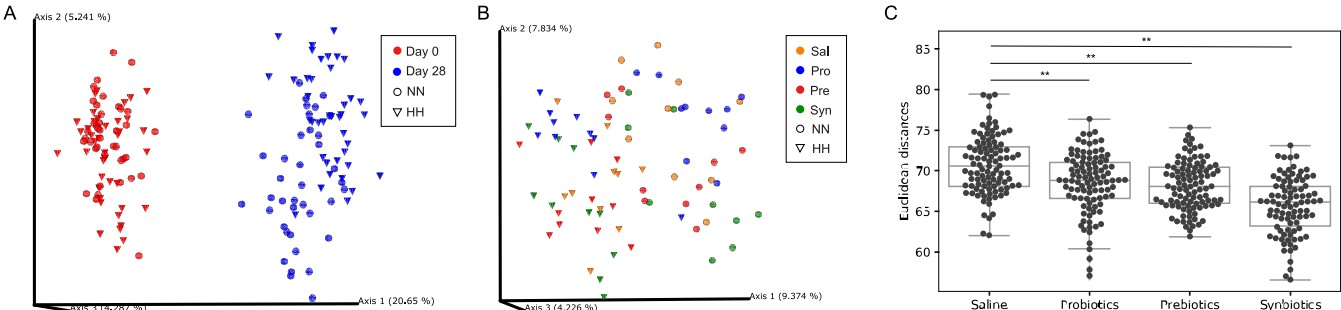

**FIG 2** The treatments ameliorated the effect of hypobaric hypoxia on gut microbiota. (A) Principal Coordinate Analysis (PCoA) of unweighted UniFrac distances of all samples. Samples are colored by day. (B) PCoA of unweigted UniFrac on day 28. Samples are colored by different treatments. Cone shape represents NN samples; sphere shape represents HH samples. Samples were clustered by shape rather than color, indicating that hypobaric hypoxia affected the overall microbiome more than treatments. (C) Euclidean distances between NN and HH samples were reduce in three treatments compared to saline group. ** $P < 0.01$.

diversity distances between NN and HH samples in the treatment groups were narrowed compared to those in the saline control group (Fig. 2C). And this improvement tends to be more prominent in synbiotics group than those in probiotics and prebiotics groups.

Next, we identified a set of individual microbes whose abundances were significantly altered by hypobaric hypoxia. Most of these microbes belong to family *Ruminococcaceae* and *Lachnospiraceae* (Fig. 3A, Table S2). The treatments reversed some of these alterations. For example, the log-ratios of ASVs of family *Ruminococcaceae* and genus *Oscillospira* were sharply reduced to NN level with the treatment of probiotics, prebiotics, or synbiotics (Fig. 3A). In the parallel study, we found hypobaric hypoxia exposure altered the abundance of genera *Parabacteroides*, *Alistipes*, *Prevotella*, *Lachnospira*, and *Lactococcus* as well as *Bacteroides* to *Prevotella* ratio. Here, we also looked into these taxa on genus level and observed that the treatments attenuated these alterations to variating degrees, except for *Lachnospira* (Fig. 3B). Taken together, these results suggest that hypobaric hypoxia challenge disrupts normal gut microbiota composition, and this disruption can be considerably alleviated by administration of probiotics, prebiotics, and synbiotics.

**Alterations of fecal SCFA and BA profiles.** SCFAs and BAs are 2 major types of active small molecules produced by gut microbiota that can affect host physiology (36). We profiled 9 SCFAs and 15 BAs in stool collected on both day 0 and day 28 with high performance liquid chromatography-mass spectrometry. Resembling the gut microbiota result, the overall fecal SCFA (Fig. 4A) and BA profiles (Fig. 4B) were both primarily driven by rat age. Mantel test also showed the correlations between day 0 and day 28 were not significant (SCFAs p-val = 0.73, BAs p-val = 0.40). Therefore, we applied similar log-ratio transformation on the fecal SCFA and BA data for downstream analyses. Unlike the gut microbiota result, PERMANOVA analysis showed that there was no statistically significant difference between NN and HH rats of saline treated control group for either SCFA or BA profile. However, the Euclidean distances of fecal SCFA profiles between NN and HH rats were decreased in the prebiotics treated group (Fig. 4C) and similarly for fecal BA profiles in the prebiotics and synbiotics treated group (Fig. 4D). This suggests that prebiotics and synbiotics might be able to offset the effect of HH on fecal SCFA and BA profiles, even though such HH effect was not statistically significant, probably due to small effect size and sample size.

We further identified three individual metabolites that were differentially abundant between NN and HH saline rats. All of them are SCFAs: 2-methylbutyric acid, isovaleric acid, and isobutyric acid. However, none of these altered SCFAs could be reversed by the treatments (Fig. 4E).

**Effect of treatments on plasma metabolome and cytokines.** By comparing metabolite profiles of germfree rodents with those of conventional (37–40), it is clearly demonstrated that a significantly large number of metabolites in the host blood arise from gut microbiome. Recent metabolomic studies revealed strong associations between gut microbial derived or host endogenous metabolites and various diseases (41). Therefore, to investigate the shift of plasma metabolome in association with hypobaric hypoxia and treatments, we quantified

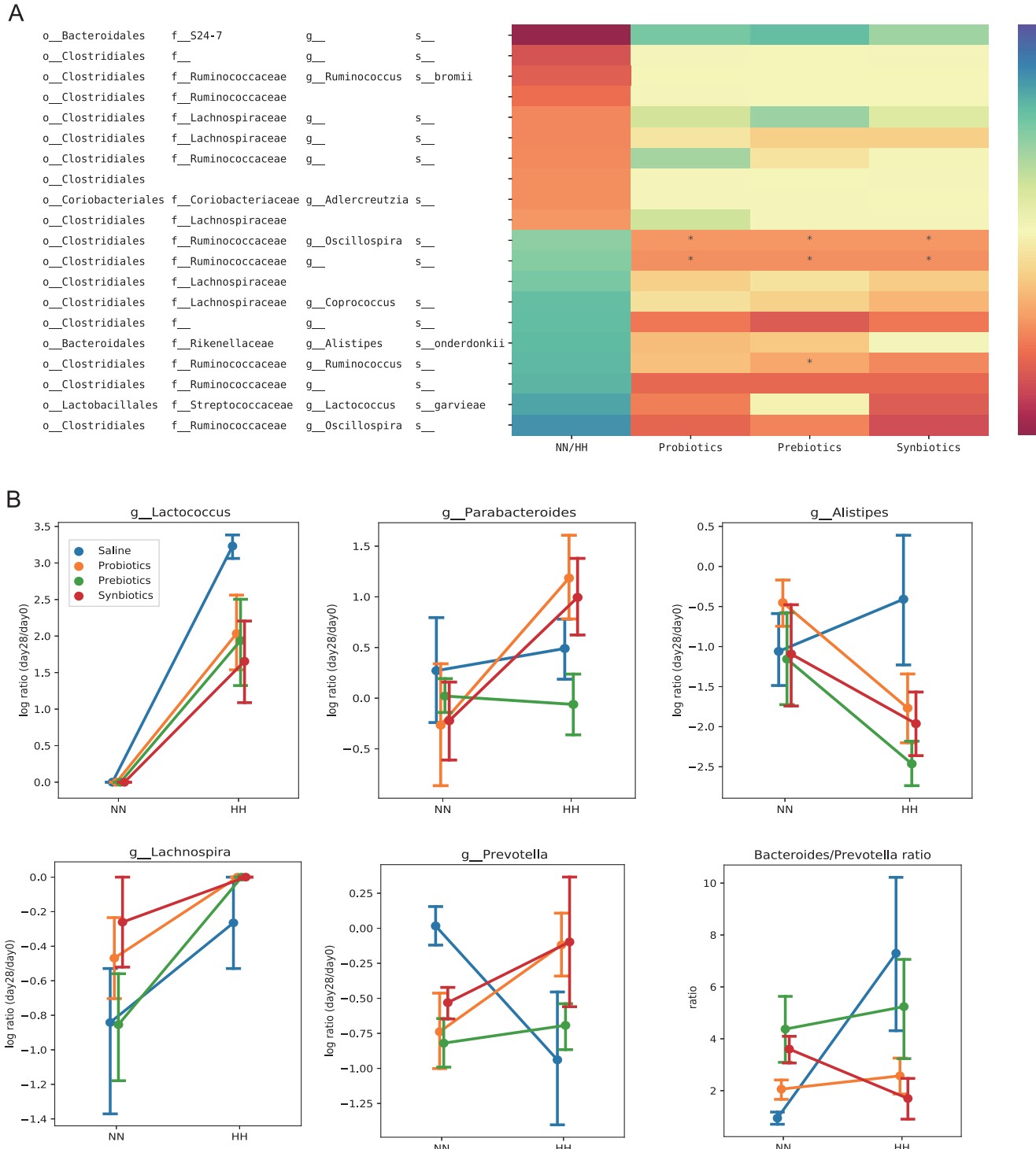

**FIG 3** Heatmap of the differential abundant microbes affected by hypobaric hypoxia. (A) Linear model (ASV ~ NN/HH environment + probiotics + prebiotics + synbiotics) was used to estimate the impact of the factors to the abundance of each ASV. The linear model coefficients of NN/HH, probiotics, prebiotics, synbiotics were visualized in the four columns of heatmap. Each row represents a ASV with significant impact of HH environment. Darker color indicates stronger impact of the factor. The heatmap shows that the treatments reversed the effect of hypobaric hypoxia on most of the ASVs to a degree. Asterisk (*) indicates False discovery rate (FDR) < 0.1. (B) Interaction plots of genus *Lactococcus*, *Parabacteroides*, *Alistipes*, *Lachnospira*, *Prevotella* and the ratio of *Bacteroides* to *Prevotella*. If a treatment line is unparallel to the saline line, it indicates there is interaction effects between the treatment and HH environment. Treatments alleviated HH induced changes in these genera except *Lachnospira*.

plasma metabolites, including SCFAs, other medium- and long-chain free fatty acids (FFAs), BAs, amino acids (AAs), neurotransmitters (NTs) and cytokines on day 28. PERMANOVA based on Aitchison distance (42) showed that hypobaric hypoxia challenge induced overall profile changes of plasma SCFAs, BAs, AAs, FFAs, and NTs (all p-val < 0.01) but not cytokines

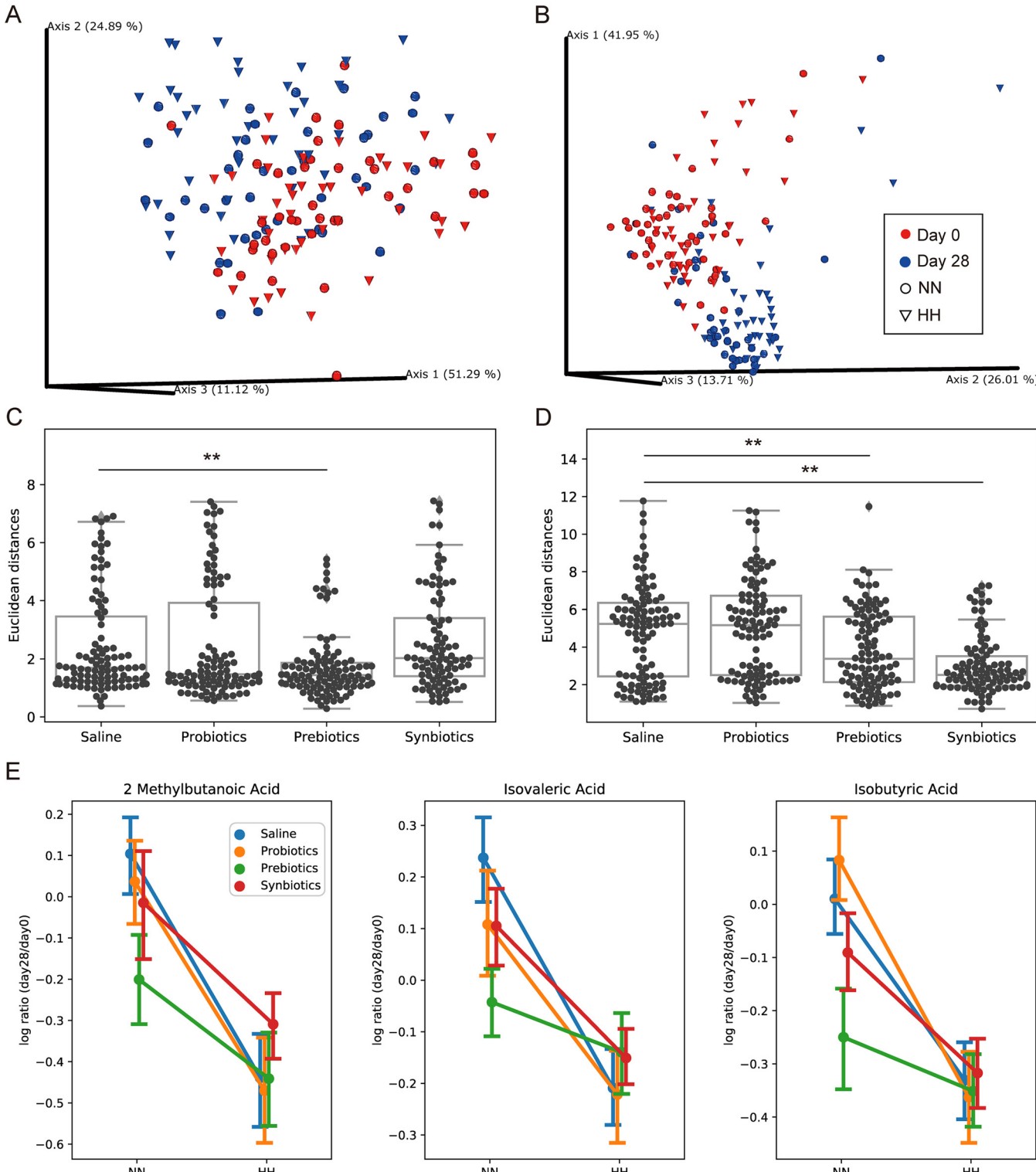

**FIG 4** Alterations of fecal SCFA and BA profiles. (A, B) PCoA of Aitchison distances showed age affected the overall composition of fecal SCFAs (A) and BAs (B). Samples are colored by day. Cone shape represents NN samples; sphere shape represents HH samples. (C, D) Euclidean distances between NN and HH samlpes in each treatment. Prebiotics reduced the distances in SCFAs (C) and synbiotics reduced the distances in BAs (D). (E) Interaction plots of differential fecal SCFAs. Treatments could not reverse the expression of these SCFAs. **, $P < 0.01$.

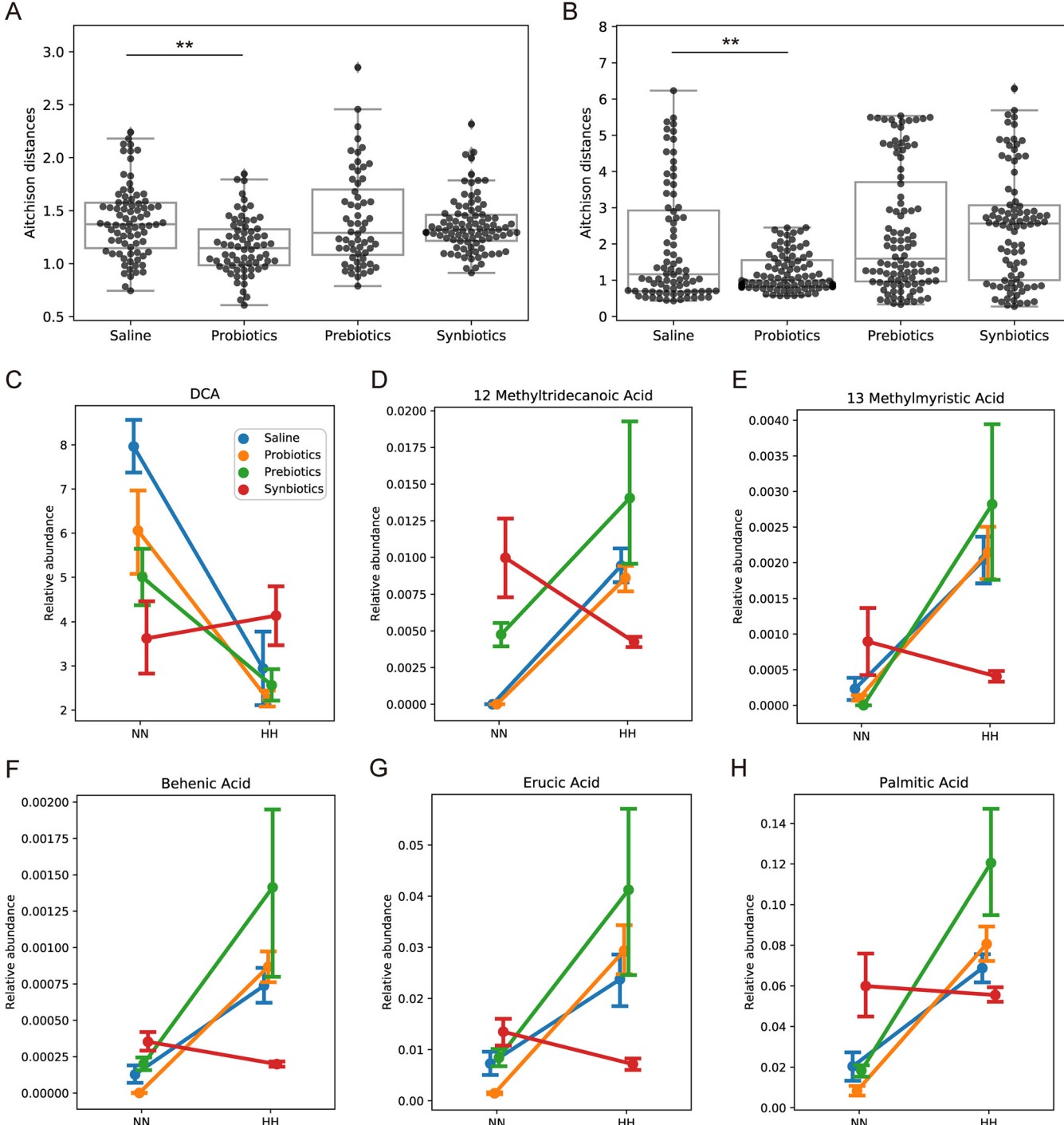

**FIG 5** Effect of treatments on plasma metabolome and cytokines. (A, B) Aitchison distances between NN and HH samples was reduced by probiotics treatment in AAs (A) and NTs (B). **, $P < 0.01$. (C–H) Interaction plot of two-way ANOVA. If a treatment line is unparallel to a treatment line, it indicates there is interaction effects between the treatment and HH environment. The relative abundance of DCA (C), 12-methyltridecanoic acid (D), 13-methylmyristic acid (E), behenic acid (F), eucic acid (G), palmitic acid (H) in HH groups was reversed especially by synbiotics. Dot represents the mean value in each group and error bar represents standard error.

(p-val = 0.062). Probiotic treatment lessened these changes for AAs (Fig. 5A) and NTs (Fig. 5B) but not for the other types of metabolites (Fig. S3A-C).

We performed two-way ANOVA analysis to examine the influence of hypobaric hypoxia and its interaction with treatments on individual metabolites. Fig. S3D showed all the metabolites that were statistically significantly altered by the prolonged hypobaric hypoxia challenge alone, including kynurenine, kynurenic acid, and glutamate. Among them, some metabolites

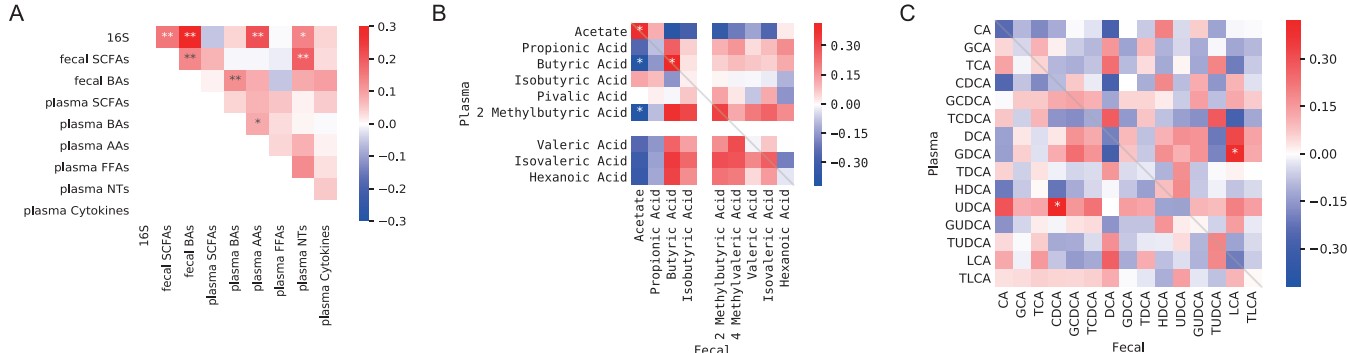

**FIG 6** Correlation between the multi-omic profiles. (A) Correlation between microbiome, fecal SCFAs, fecal BA, plasma SCFAs, plasma BAs, plasma AAs, plasma FFAs, plasma NTs, plasma cytokines was analyzed by Mantel test. (B, C) Visualization of Spearman correlation between fecal SCFAs and plasma SCFAs (B), and between fecal BAs and plasma BAs (C). Dark red color indicates strong positive correlation, dark blue indicates strong negative correlation. Asterisk indicates the matched pairs were statistically significantly associated with each other. One star (*) indicates FDR < 0.05, two stars (**) indicate FDR < 0.01.

were observed with statistically significant interaction effects of HH and the treatments. The levels of these HH-altered metabolites (mostly BAs and FFAs) were statistically significantly restored by treatments, most prominently seen in synbiotics-treated group (Fig. 5C to H).

**Correlation between the multi-omic profiles.** For each pair of collected data sets, we performed Mantel test to look for correlations between their overall compositions (Fig. 6A). Unsurprisingly, we found strong correlations among gut microbiota, fecal (but not plasma) SCFA and BA profiles. Gut microbiota was also correlated with amino acids and neurotransmitters, reflecting the impact of gut microbes on amino acid metabolism and neuroactive molecule production. Looking at individual SCFA, we observed that acetate levels in plasma and stool was positively correlated, but they were negatively correlated with most of other SCFAs. Fecal butyric acid was significantly positively correlated with plasma butyric acid, and moderately correlated with 2-methylbutyric acid, isovaleric acid, and hexonoic acid (Fig. 6B). Overall, fecal BAs were correlated with plasma BAs (Fig. 6A), probably because majority of BAs in the gut are reabsorbed back via enterohepatic circulation. Most individual correlations between plasma and fecal BAs were not statistically significant except for plasma UDCA versus fecal CDCA and plasma GDCA versus fecal LCA (Fig. 6C).

We performed pairwise correlation analysis between fecal microbes and fecal/plasma metabolites. After multiple hypotheses correction, we listed all the significant correlations as in Fig. 7 (FDR < 0.01, cor coefficient > 0.5). We observed that the microbes and metabolites could be roughly divided into two correlation patterns after sorting the rows and columns according to their correlation coefficients. Specifically, three microbes of family Christensenellaceae, which was reported as highly heritable and depleted to obesity and metabolic syndrome [43], were all negatively correlated with Phe, Glu, and Cholesterol.

## DISCUSSION AND CONCLUSIONS

Hypobaric hypoxia is the main factor responsible for the development of high-altitude illness. Complex interactions between human genetics and the environment factors contribute to individual susceptibility to hypoxia-induced high-altitude illness. Its most effective treatments are descent of altitude, supplemental oxygen, and medical therapies such as administration of acetazolamide. Recently, emerging evidence, including our parallel study, has demonstrated that hypobaric hypoxia has a profound influence on gut microbiome. However, it is still inconclusive whether hypoxia-induced syndrome and gut microbiome shift is merely an association or there is causal link between them. If gut microbiome dysbiosis contributes to the manifestation of high-altitude diseases, gut microbiome-targeted intervention may serve as an effective treatment.

In the present study, we administered probiotics, prebiotics, and synbiotics to hypobaric hypoxia challenged rats. The gavaged probiotics were verified via qPCR experiment. These treatments changed the HH-induced gut microbiota toward the direction of normal composition, although their effects are limited on overall gut microbial profile, compared

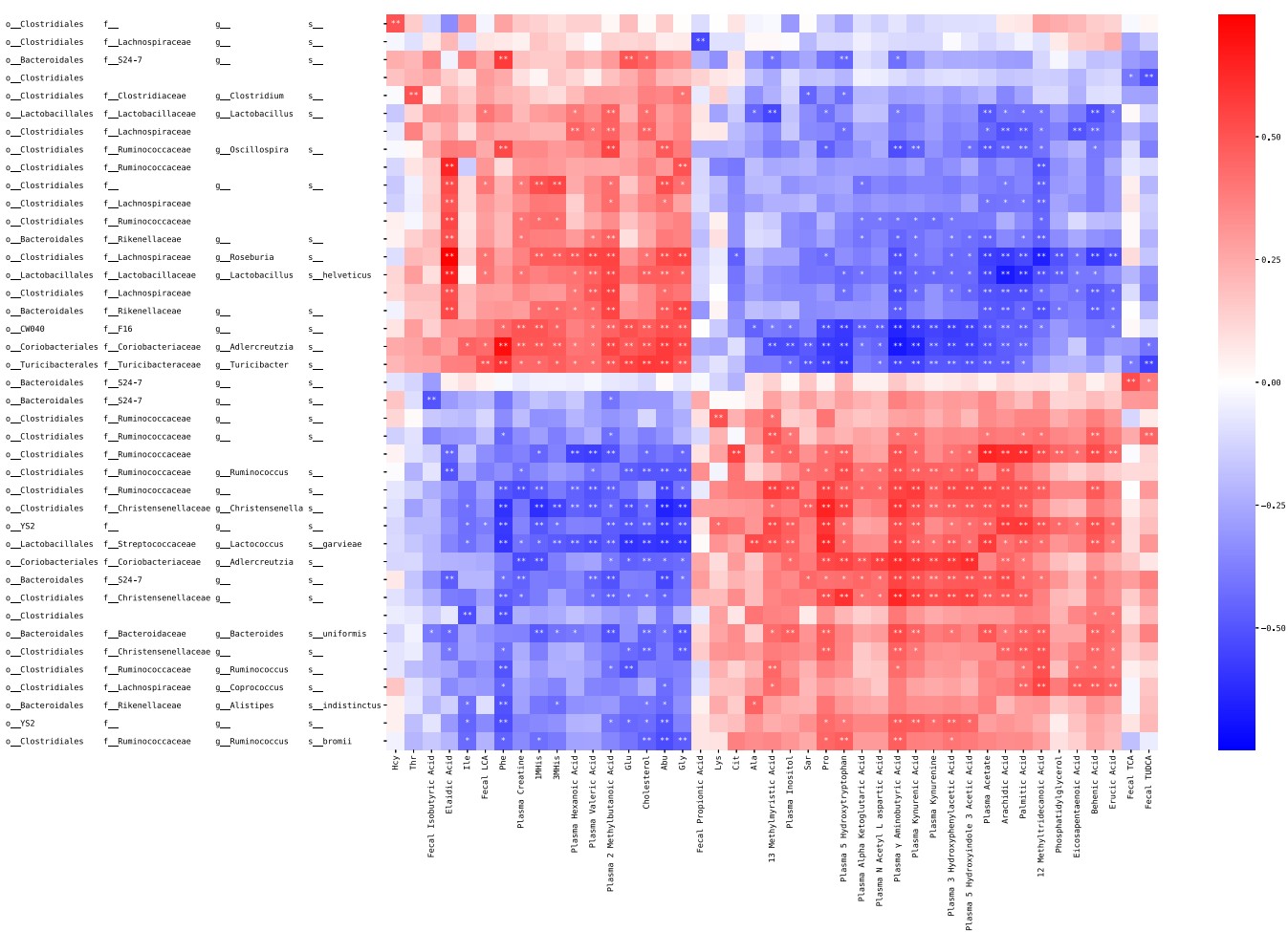

**FIG 7** Pairwise correlation between fecal microbes and fecal and plasma metabolites. Each row is an ASV, each column is a metabolite. Dark red color indicates strong positive correlation, dark blue indicates strong negative correlation. Asterisk indicates the matched pairs were statistically significantly associated with each other. One star (*) indicates FDR < 0.05, two stars (**) indicate FDR < 0.01.

to the age and hypobaric hypoxia environment. We identified bacteria that were significantly altered by HH. Interestingly, their abundances recovered toward the NN level with three treatments.

Hypobaric hypoxia induces substantial metabolic reprogramming (44, 45). To comprehensively profile these metabolic shifts, we measured a large number of fecal and plasma metabolites, probably the largest to our knowledge. We identified a set of plasma short-chain fatty acids, other free fatty acids, bile acids, amino acids, and neuro active molecules were altered in HH rats. Notably, the levels of three medium-long-chain FFAs in plasma were significantly increased in HH-challenged rats, they were palmitic, erucic, and behenic acid (Fig. 5F and G). The levels of plasma FFAs were reported be associated with the heart diseases (46, 47). Previous studies consistently demonstrated an increase in plasma palmitic acid (PA) under hypoxia environment (13, 45, 48). Elevated plasma PA was associated the development of cardiovascular diseases (49). PA caused a decrease in insulin secretion and viability of islet cell in *ex vivo* (48). Plasma erucic acid was able to cross the blood-brain barrier and was incorporated into specific lipid metabolism via chain shortening (50).

Specifically, a sharp decrease of Glu/Pro and Gln/Pro ratios were observed after hypobaric hypoxia exposure, while the Glu/Gln ratio was also decreased, to a lesser degree (Fig. S4). Proline plays an important regulatory role in cancer and is found as an important metabolite during adaption to hypoxia (51–53). In human, Gln can be converted to Glu by glutaminase; Glu can be catalyzed from Gln by glutamine synthetase. Conversion of Gln and Glu to Pro via pyrroline 5-carboxylate (P5C) synthase is found enhanced in response to hypoxia in carcinoma

Microbiology
Spectrum

(54), which agrees with our observation of their ratio alterations. This suggests that high-altitude animal model may share some similarities with pathophysiological process of hypoxia in tumorigenesis. However, our treatments of probiotics, prebiotics, and synbiotics were not able to restore this particular amino acid metabolic reprogramming except for the mild alleviation on Glu/Gln ratio (Fig. S4C). Furthermore, we cross-correlated these multi-omic profiles and found that fecal and plasma metabolomics and gut microbiome data sets showed a certain degree of correlation, reflecting the effect of the microbiome on metabolic activity. Many individual SCFAs and BAs did not have correlated levels in plasma and stool, implying that either the metabolite exchanges of blood and gut are actively regulated, or their levels are significantly perturbed by other metabolic processes unaccounted here.

In conclusion, the administration of probiotics, prebiotics, and synbiotics significantly attenuated cardiac hypertrophy inflicted by prolonged hypobaric hypoxia exposure. The hypobaric hypoxia-induced gut microbiome shift, as well as metabolome alterations, was also ameliorated by the interventions, suggesting that gut microbiome modulation can serve as a therapeutic treatment for high-altitude diseases.

## MATERIALS AND METHODS

**Animals and study design.** Eighty specific-pathogen free male Wistar rats were obtained from Beijing Vital River Laboratory Animal Technology Co., Ltd. (Beijing, China). All rats were raised in homogeneous conditions. As shown in Fig. S1, after a 6-day adaptive period, the rats were randomly divided to normobaric normoxia (NN) and hypobaric hypoxia (HH) group. NN rats were kept at sea level; HH rats were transported into a chamber to simulate the altitude at 5,000 m for continuous 28 days. HH rats were maintained in sustained hypoxia for the first week. To guarantee the safety of researchers and quality of HH, we changed the housing condition to intermittent hypobaric hypoxia ($>$20 hypoxic hours/per day) for the next 3 weeks. Our study aimed to identify the effect of probiotics/prebiotics via gut microbiome, thus the rats in each environment were gavaged with saline, probiotics, prebiotics, or synbiotics everyday (10 rats/group).

**Reagents and treatments.** The commercial probiotics was obtained from Beijing Hi-LongCare MedPharmaicine & Technology Co., Ltd. (Beijing, China). According to the company's claim, it contains 3 *Bifidobacterium* strains (*B. animalis subsp. lactis V9*, *B. longum KT-L9*, *B. adolescentis KT-A8*) and 6 *Lactobacillus* strains (*L. casei Zhang, L. plantarum P-8, L. paracasei KT-P6, L. rhamnosus M9, L. acidophilus KT-A1, L. helveticus H9*). Prebiotics was provided by Beijing Future Science & Technology Development Co., Ltd., which mainly consists of polydextrose, galactose, and inulin. Synbiotics is a mixture of probiotics and prebiotics. All reagents were dissolved in physiological saline (0.9% sodium chloride). All the medications were given by gavage at the daily volume of 3 ml saline per rat. The probiotics was administered at a dose of $5 \times 10^9$ CFU (CFU)/rat, the prebiotics was administered at a dose of 1g/rat.

**Sample collection and processing.** Fecal samples were collected at 3 days before transferring to the chamber (this sample marked as day 0 for convenience) and day 28. Heart and plasma samples were collected at the end of experiments. Heart samples were photographed.

**DNA extraction and quantitative PCR amplification.** For heart tissues, total RNA was extracted using TRIzol reagent and reverse-transcribed to cDNA using SuperScript II reverse transcriptase. The marker genes of cardiac hypertrophy at the apex of the heart were quantized via real-time PCR as described previously (55). The primers were listed in the supplemental materials (Table S3).

To verify the probiotics in fecal samples, seven primer pairs for identification of probiotics were selected from previous reports using quantitative PCR method (Table S3). Bacterial genomic DNA from the rat stools was extracted by the QIAam Fast DNA Stool minikit (Qiagen, Germany, cat. no. 51604) according to the manufacturer's instructions. Resulting DNA concentrations were tested using a Nanodrop 2000 spectrophotometer (Thermo Fisher Scientific, USA), and 10 ng/$\mu$l of each sample template was directly applied for amplification with the Roche LightCycler 480 instrument and SYBR greenIMaster mix (Roche). Fluorescence detection was performed at the end of the amplification step. The relative expression of target gene to 16S rRNA variable region gene was calculated.

**16S amplicon sequencing.** For fecal samples, DNA extraction and purification were performed using QIAamp Fast DNA Stool minikit. Amplicon library for bidirectional sequencing on Illumina MiSeq PE250 platform was constructed using primers 341F 5'-CCTACGGGRSGCAGCAG-3', 806R 5'-GGACTACVVGGGTATCTAATC-3' targeted across 16S rRNA genes V3-V4 regions.

**Fecal metabolites determination.** Targeted metabolomics in feces and plasma were analyzed at Longsee Biomedical Corporation by UHPLC/MS system. Stool weighing 100 mg was added with added with 400 $\mu$l extract of methanol-acetonitrile-water (2:2:1, vol/vol/vol), vortexed and centrifuged at 17,000 g for 15 min, and the supernatant was collected for further measurement.

For BAs test, after filtering through PVDF membrane (0.22 $\mu$m), the supernatants were subjected to metabolomics profiling by a Nexera ultrahigh performance liquid chromatography (UHPLC, LC-30, Shimadzu) coupled to a triple quadrupole mass spectrometer (MS, LCMS-8050, Shimadzu) system in multiple reaction monitoring (MRM) mode using both positive and negative electrospray ionization (ESI). Chromatographic separation was performed on a column of ACQUITY UPLC HSS T3 (1.8 $\mu$m, 2.1 mm $\times$ 100 mm, Waters, Milliford, MA, USA). The column temperature was set at 40°C. Single injection at volume of 1 $\mu$l was loaded on the trap. The mobile phase was consisted of solvent A (5 mM ammonium acetate in water) and solvent B (acetonitrile).

The gradient program was as follows: 0 min, 35% B; 2 min, 35% B; 3 min, 41% B; 4 min, 45% B; 5.3 min, 45% B; 5.5 min, 80% B; 7.5 min, 80% B; 8.7 min, 35% B; 10 min, 35% B; and 10 min, stop.

For SCFAs measurement, the supernatant (100 $\mu$l) was treated with 12C-aniline (33.5 $\mu$l, 21.47 mmol/liter) and 1-ethyl-3-(3-dimethylaminopropyl) carbodiimide solution (2 $\mu$l, 175 mmol/liter) for derivatization for 2 h, followed by quenching with $\beta$-mercaptoethanol (0.8 $\mu$l, 500 mmol/liter) and succinic acid (6 $\mu$l, 150 mmol/liter) for 2 h, and then adding with the 50% methanol to yield the dilution at the final volume of 300 $\mu$l. The mixture was subjected to metabolomics profiling by UHPLC-MS. The mobile phase was consisted of solvent A (formic acid: water, 1:1000) and solvent B (2-propanol: water, 40:60). The gradient program was as follows: 0 min, 45% B; 2 min, 45% B; 6 min, 55% B;17 min, 57% B; 17.2 min, 45% B; 23 min, 45% B; 23 min, stop.

Source conditions were optimized: nebulizing gas flow 3 L/min, heating gas flow 13 L/min, drying gas flow 7 L/min, interface temperature 350℃, DL temperature 150℃, and heat block temperature 400℃. A flow rate of 0.4 ml/min was both utilized in pump. Data were all obtained by using the LabSolutions workstation.

**Plasma metabolites determination.** The instruments and analytical conditions were equal to those of fecal metabolites determinations except for sample preparations. For BAs test, plasma was added with extract of methanol-acetonitrile (1:1, vol/vol) to facilitate protein precipitation. After vertexing and centrifuging, the supernatant was collected followed by vacuum drying, and then re-dissolved with methanol-water (1:1, vol/vol). For SCFAs measurement, sample was mixed with extract of methanol-acetonitrile, followed by derivatization and quenching using above-mentioned methods.

For neurotransmitters (NTs), plasma sample was added with extract of methanol-acetonitrile (1:1, vol/vol) to facilitate protein precipitation. After vortexing and centrifuging, the supernatant was collected followed by vacuum drying, and then re-dissolved with methanol-water (1:1, vol/vol). For free fatty acids (FFAs) quantification, each plasma sample was mixed with isopropanol-hexane (80:20, vol/vol), treating with 2% phosphoric acid, and followed by shaking and extraction with hexane-water (4:3, vol/vol). The supernatant was collected after centrifugation, vacuum drying, and re-dissolved with methanol-water (1:1, vol/vol). Finally, all prepared samples were subjected to metabolomics profiling by UHPLC/MS system.

For plasma amino acids measurement, each sample (40 $\mu$l) was deproteinized with 10% sulfosalicylic acid, and then added with internal standards (10 $\mu$l) and water (120 $\mu$l). After vortexing and centrifuging at 4,000 rpm for 10 min at 4℃, 1 $\mu$l of supernatant was injected into the LC/MS system for analysis at The Beijing Genomics Institute (BGI)-Shenzhen (Shenzhen, China). The analysis system consisted of a Transcend II ultrahigh performance liquid chromatography system (ThermoFisher Scientific, PA, USA) and an QTRAP 5500 mass spectrometer system (AB SCIEX, MA, USA) in MRM mode with positive ESI source. Data acquisition and calculation was performed by MultiQuant software (SCIEX, Framingham, MA). Chromatographic separation was achieved on a column of ACQUITY UPLC HSS T3 (1.8 $\mu$m, 2.1 mm $\times$ 100 mm, Waters, Milliford, MA, USA). Mobile phase A was water containing 0.1% formic acid and 0.05% heptafluorobutyric acid, and mobile phase B was acetonitrile-0.1% formic acid and 0.05% heptafluorobutyric acid. A mobile phase gradient was applied at a flow rate of 0.5 ml/min. The gradient elution was initial, 2% B; 0.5 min, 2% B; 1.5 min, 10% B; 3.5 min, 35% B; 3.6 min, 95% B; 5.0 min, 95% B; 5.01 min, 2% B; and 6.5 min, 2% B. For plasma cytokines measurement, samples were analyzed using the Bio-Plex 200 analyzer at Laizee Biotech Co, Ltd. (Beijing, China). A Rat Cytokine & Chemokine Panel (22 plex, ProcartaPlex Multiplex Immunoassays, eBioscience, USA) was used to test plasma levels of cytokines according to the manufacturer's instructions. A panel of cytokines/chemokines listed as follow: IL-1$\alpha$, G-CSF, IL-10, IL-17A, IL-1$\beta$, IL-6, TNF-$\alpha$, GM-CSF, IL-4, IFN-$\gamma$, IL-2, IL-5, IL-13, IL-12p70, GRO$\alpha$, MCP-1, MIP-1$\alpha$, Rantes, Eotaxin, MIP-2, IP-10, and MCP-3.

Abbreviations for metabolites were as follows: CA: cholic acid; GCA: glycocholic acid; TCA: taurocholic acid; CDCA: chenodeoxycholic acid; GCDCA: glycochenodeoxycholic acid; TCDCA: taurochenodeoxycholic acid; DCA: deoxycholic acid; GDCA: glycodeoxycholic acid; TDCA: taurodeoxycholic acid; HDCA: hyodeoxycholic acid; UDCA: ursodeoxycholic acid; GUDCA: glycoursodeoxycholic acid; TUDCA: tauroursodeoxycholic acid; LCA: lithocholic acid; TLCA: taurolithocholic acid; Arg: arginine; His: histidine; Ile: isoleucine; Leu: leucine; Lys: lysine; Met: methionine; Phe: phenylalanine; Thr: threonine; Trp: tryptophan; Val: valine; Glu: gutamic acid; Orn: ornithine; Cit: citrulline; Asn: asparagine; Gln: glutamine; Aad: $\alpha$-aminohexanoic diacid; Asp: aspartic acid; Cys: cystine; Cth: cystathionine, D-(P); Hcy: homocystine; Ser: serine; Tyr: tyrosine; Gly: glycine; Tau: taurine; Abu: $\alpha$-aminobutyric acid; Pro: proline; Ala: alanine; Hcit: homocitrulline; BAla: $\beta$-alanine; Car: carnosine; Sar: sarcosine; 1MHis: 1-methyl-L-histidine; 3MHis: 3-methyl-histidine; Hyp: hydroxyproline; Pser: phosphoserine; Hyl: $\gamma$-hydroxylysine; PEtN: phosphorylethanolamine.

**Analysis of 16S amplicon microbiome data.** Analysis of the 16S rRNA gene sequences were performed with Quantitative Insights Into Microbial Ecology version 2 (QIIME2, version 2019.4) (56). In brief, paired-end raw reads were first joined. QIIME2 deblur plugin was used to quality control and construct a high-quality amplicon sequence variant data (57). An average of 7,348 reads per sample were obtained after this step. Taxonomy assigned against Greengenes database using QIIME2 q2-feature-classifier plugin. ASV table was rarefied to 4,515 sequences per sample to remove the bias of uneven sequence sampling. Alpha and beta diversity metrics were calculated by q2-diversity plugin. Emperor plugin was used to visualize Principal Coordinates Analysis (PCoA) plots. To minimize the influence of age effect, we calculated the log ratio of relative abundance on day 28 over day 0 and applied Euclidian metric to compute beta diversity distances (as UniFrac metric does not support negative values in the log-ratios). The distances between NN and HH samples were computed to assess the effects of treatments on microbiome. Permutational multivariate analysis of variance (PERMANOVA) was computed to evaluate significant differences between groups using python scikit-bio library.

**Analysis of metabolome data.** We applied total sum normalization for every metabolomic profiles. For fecal SCFAs and BAs, we applied log-ratio transformation of day 28 over day 0, similarly to what was done for 16S amplicon data. Beta diversity and PCoA plots were computed similarly as described above.

**Statistical analyses of microbiome and metabolome data.** Considering the age effect, we therefore combined log-ratio transformation and linear model to distinguish the differential ASVs between NN and HH in saline group, followed by multiple hypothesis testing correction with Benjamini-Hochberg method. The model was fitted with formula using R stat package:

$$log-ratio \sim NN/HH \ environment \ + \ probiotics \ + \ prebiotics \ + \ synbiotics$$

where $log-ratio = \log_{10}$ the relative abundance at day 28 / the relative abundance at day 0) . 20 ASVs were discovered with the significant criterion of False discovery rate (FDR) $<$ 0.1. Differential fecal metabolites were detected by the same method.

For plasma metabolites, we first employed $t$ test (scipy.stats.ttest_ind) to identify the differential metabolites between NN and HH groups in saline group of samples. To estimate the treatment effects on these HH affected metabolites, we applied two-way ANOVA (statsmodels.stats.anova.anova_lm) to unravel significant interaction effects between treatment and HH. Significant interaction indicates that a treatment influences the effect of HH. Omega squared was calculated to quantify the effect size of this interaction of treatments on HH (58).

Mantel test (skbio.stats.distance.mantel) was used to compute correlation between distance matrices of every omics profiles. Pairwise correlations between microbes and metabolites were calculated using Spearman's rank (scipy.stats.spearmanr) followed by FDR correction.

**Data availability.** The raw sequencing data are available from the NCBI Sequence Read Archive with accession numbers PRJNA715614 and PRJNA715837.

## SUPPLEMENTAL MATERIAL

Supplemental material is available online only.

**SUPPLEMENTAL FILE 1**, PDF file, 1.9 MB.

## ACKNOWLEDGMENTS

All animal procedures were carried out in accordance with the Declaration of the National Institutes of Health Guide and Use of Laboratory Animals and approved by the animal care and use committee of Beijing Institute of Microbiology and Epidemiology (No. IACUC-DWZX-2017-005).

This research was supported by the National Natural Science Foundation of China (grant no. 81790632, 31970863, and 31970088) and the National Key Technology Research and Development Program of the Ministry of Science and Technology of China (grant no. 2020YFA0509600).

We declare that we have no conflicts of interest.

Z.P., Z.H., N.H., and X.Z. did the experiments. H.P. and Q.G. performed the hypobaric chamber. Q.W. and H.Z. provided probiotics. B.J.Y.L. provided prebiotics. Y.H., Y.L., and Z.Z.X. analyzed the data. Y.H., Z.P., Y.B., and Z.Z.X. wrote the manuscript. Z.Z.X., Y.B., and R.Y. designed the experiments and contributed to manuscript revisions. R.Y. provided overall direction for the experiments.

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
