## [Reviewer comments · Microbiology Spectrum]

Microbiology Spectrum

Gut microbiome-targeted modulations regulate metabolic profiles and alleviate altitude-related cardiac hypertrophy in rats

Zhenjiang Xu, Yichen Hu, Zhiyuan Pan, Zongyu Huang, Yan Li, Ni Han, Xiaomei Zhuang, Hui Peng, quansheng gao, Qing Wang, B.J. Yang Lee, Heping Zhang, Ruifu Yang, and Yujing Bi

Corresponding Author(s): Zhenjiang Xu, Nanchang University

Review Timeline:

Submission Date:	July 26, 2021
Editorial Decision:	September 19, 2021
Revision Received:	November 10, 2021
Accepted:	November 30, 2021

Editor: Jennifer Auchtung

Reviewer(s): Disclosure of reviewer identity is with reference to reviewer comments included in decision letter(s). The following individuals involved in review of your submission have agreed to reveal their identity: Mustafa Özçam (Reviewer #1); Jie Hong (Reviewer #2)

Transaction Report:

DOI: <https://doi.org/10.1128/Spectrum.01053-21>

September 19, 2021

Dr. Zhenjiang Zech Xu
Nanchang University
State Key Laboratory of Food Science and Technology
Nanchang, Jiangxi
China

Re: Spectrum01053-21 (Gut microbiome-targeted modulations regulate metabolic profiles and alleviate altitude-related cardiac hypertrophy in rats)

Dear Dr. Zhenjiang Zech Xu:

Thank you for submitting your reviews. As you can see, both reviewers had comments and suggestions that should be addressed in your revised submission. When submitting the revised version of your paper, please provide (1) point-by-point responses to the issues raised by the reviewers as file type "Response to Reviewers," not in your cover letter, and (2) a PDF file that indicates the changes from the original submission (by highlighting or underlining the changes) as file type "Marked Up Manuscript - For Review Only". Please use this link to submit your revised manuscript - we strongly recommend that you submit your paper within the next 60 days or reach out to me. Detailed information on submitting your revised paper are below.

Link Not Available

Sincerely,

Jennifer Auchtung

Journals Department
Reviewer comments:

Reviewer #1 (Comments for the Author):

The manuscript "Gut microbiome-targeted modulations regulate metabolic profiles and alleviate altitude-related cardiac hypertrophy in rats" by the authors Hu, and Pan et al. describes the effect of probiotics, prebiotics, and symbiotics in hypobaric hypoxia severity using a rat model. Due to the lack of research in 16S rRNA amplicon sequencing and fecal and plasma metabolome combination, the manuscript presents exciting findings, including prebiotics increasing neurotransmitter levels, which might be further investigated in another study. The manuscript is written very nicely and is easy to follow. I have some comments below.

Major Comments

1. Interestingly, HH increased three saturated medium-long chain fatty acids (palmitic, erucic, and behenic) levels, and symbiotics reduced their levels. Could the authors please explain the role of these fatty acids in hypobaric hypoxia development and microbiota dysbiosis?

1. The authors state that they could not identify the probiotics in the microbiota data after 28 days of treatment but also stated

that the changes in amino acid levels in plasma are associated with the microbiota (lines 273-275). How do authors explain this?

1. The authors refer to multiple figures with one sentence, such as (Fig 1A-C). Please refer to each figure individually in the result section, such as Figure 1A and Figure 1B instead of Figure 1A-C.

1. Please explain how were the prebiotics and probiotics used in this study were chosen?

Minor Comments

Line 30-31. Please clarify if the disruption of gut microbiota was ameliorated or partially ameliorated? Although the abstract reads very strong, the result section is written with more moderate language.

Line 69-70. Please reconsider describing Faecalibacterium, Lachnospiraceae as health-promoting microbes as it reads like a vague statement without a solid scientific basis.

Line 76. I would avoid using aberrant microbiota as it is not scientifically classified.

Line 77-78. Reference 21 (Bartolomaeus et al.) does not state any probiotic usage. They only used propionate. Please clarify if I am missing something.

Line 82. Please consider making the following change "gut microbiota is a"> "gut microbiota might be a."

Line 89. What do you mean by had a greater impact? Compared to what other variables?

Line 92. Do you mean modulating dysbiosis in gut microbiota instead of "gut dysbiosis"? Or please explain what do you mean by "gut dysbiosis."

Figure 1B and C. Heat > Heart.

Figure 1B and C. Could the author provide individual data points for these figures as they did in Figures 1D and E?

Line 720-722. Why do authors use two different symbols for the same p-value, such as ### and ** for $p < 0.001$?

Line 122. Please consider removing "mainly."

Line 722. Please provide details of what statistical tests were used to compare each experimental group in Figure 1.

Line 200. Would you please delete "statistically"?

Line 202-207. It is not clear what authors mean by diversity distances narrowed between groups and why this is considered an improvement? I am not sure if figure 2C shows this.

Line 208-209. Please explain how these microbes were determined, by, for example, which test?

Line 223. It is not clear why the authors wanted to test SCFA and BA. This may be improved by further explaining the relationship between these two metabolite groups and the HH.

Line 240. Why do authors think that it is due to the small effect and sample size? It looks like there were more than 20 data points in those experiments.

Line 255. Please cite "Aitchison distance," if applicable.

Why were neurotransmitters tested? How about those data in NN groups.

Line 311-314. Please provide this information in the related result section, as it is essential to evaluate the rest of the microbiome data. Please also explain the reason for not being able to detect the probiotics in the microbiota data is. Is that because of the caveats with sequencing, sample collection, or library preparation, or do you think the microbes were washed out from rats in 28 days?

Line 318-320. Would you please provide a citation for this statement?

Line 335. Would you please provide a citation for this statement

Reviewer #2 (Comments for the Author):

The manuscript written by Pan et al. described the dynamic changes of gut microbiota, short-chain fatty acids, and bile acids in a hypobaric hypoxia-challenged rat model for 28 days. They found the key bacteria significantly correlated with the metabolic abnormalities of SCFAs and BAs. Meanwhile, FMT moderately ameliorated cardiac hypertrophy in model rats and regulated the composition of gut microbiota. The study is interesting and the design is prudent. However, I have several comments for the study.

1. As the method illustrated, during the first 7 days, the rats were kept in a state of sustained hypoxia which mimic the rush entry into high altitude areas, and in the following 21 days, the rats being in a chronic intermittent hypobaric hypoxic state. Thus, the first 7 days even only the first day maybe take on obvious regulation of gut microbiota, whether analysis the changes of the first 7 days alone will show different results and find more difference?

2. Compared to the control group, the correlations between gut microbiome and fecal metabolites were remarkably lost in model rats. I noticed nearly all the interactions disappeared in hypoxia, and no interaction between bacteria and metabolisms. But maybe it not true. Possibly new interaction established in hypoxia out of these bacteria and metabolisms. Thus, can you build the new microbiome-metabolome interactions based on the data from hypoxia, which showed the real interaction in the hypoxia situation.

3. FMT was important to clarify the essential roles of gut microbiota from normal condition to hypoxia. Results showed abundance of some bacteria partly recovered to normal situation, how about metabolites, can they recover?

4. Composition of SCFAs and BAs in plasma between normoxia and hypoxia is obviously different as shown in Fig. S10. It looks more obvious than that in feces. But no statistical significance with Mantel test were found between gut microbiota and the

circulating metabolites at day 28. Can author explain why the obvious changes of SCFAs and BAs in plasma? Are there other influence factors?

5. The title seems need to be revised. "Cardiac hypertrophy" was only a marker (clinical manifestation, similar to decreased hemoglobin) of hypoxia challenge, but the aim of this paper was to detect the changes of gut microbiota in hypoxia, not to study cardiac hypertrophy. Thus, in the title there is no essential to emphasize "altitude-induced cardiac hypertrophy". In addition, the authors observed the gut microbiota of rats in hypoxia for 28 days, thus please revise the title into "Dynamic changes in gut microbiota and metabolites in rats during hypobaric hypoxia challenge"?

6. This manuscript still needs more revision by a native English speaker.

Staff Comments:

Preparing Revision Guidelines

Please return the manuscript within 60 days; if you cannot complete the modification within this time period, please contact me. If you do not wish to modify the manuscript and prefer to submit it to another journal, please notify me of your decision immediately so that the manuscript may be formally withdrawn from consideration by Microbiology Spectrum.

The manuscript "Gut microbiome-targeted modulations regulate metabolic profiles and alleviate altitude-related cardiac hypertrophy in rats" by the authors Hu, and Pan *et al.* describes the effect of probiotics, prebiotics, and symbiotics in hypobaric hypoxia severity using a rat model. Due to the lack of research in 16S rRNA amplicon sequencing and fecal and plasma metabolome combination, the manuscript presents exciting findings, including prebiotics increasing neurotransmitter levels, which might be further investigated in another study. The manuscript is written very nicely and is easy to follow. I have some comments below.

Major Comments

1. Interestingly, HH increased three saturated medium-long chain fatty acids (palmitic, erucic, and behenic) levels, and symbiotics reduced their levels. Could the authors please explain the role of these fatty acids in hypobaric hypoxia development and microbiota dysbiosis?
1. The authors state that they could not identify the probiotics in the microbiota data after 28 days of treatment but also stated that the changes in amino acid levels in plasma are associated with the microbiota (lines 273-275). How do authors explain this?
1. The authors refer to multiple figures with one sentence, such as (Fig 1A-C). Please refer to each figure individually in the result section, such as Figure 1A and Figure 1B instead of Figure 1A-C.
1. Please explain how were the prebiotics and probiotics used in this study were chosen?

Minor Comments

Line 30-31. Please clarify if the disruption of gut microbiota was ameliorated or partially ameliorated? Although the abstract reads very strong, the result section is written with more moderate language.

Line 69-70. Please reconsider describing Faecalibacterium, Lachnospiraceae as health-promoting microbes as it reads like a vague statement without a solid scientific basis.

Line 76. I would avoid using aberrant microbiota as it is not scientifically classified.

Line 77-78. Reference 21 (Bartolomaeus et al.) does not state any probiotic usage. They only used propionate. Please clarify if I am missing something.

Line 82. Please consider making the following change "gut microbiota is a"> "gut microbiota might be a."

Line 89. What do you mean by had a greater impact? Compared to what other variables?

Line 92. Do you mean modulating dysbiosis in gut microbiota instead of "gut dysbiosis"? Or please explain what do you mean by "gut dysbiosis."

Figure 1B and C. Heat > Heart.

Figure 1B and C. Could the author provide Individual data points for these figures as they did in Figures 1D and E?

Line 720-722. Why do authors use two different symbols for the same p-value, such as ## and ** for $p < 0.001$?

Line 122. Please consider removing "mainly."

Line 722. Please provide details of what statistical tests were used to compare each experimental group in Figure 1.

Line 200. Would you please delete "statistically"?

Line 202-207. It is not clear what authors mean by diversity distances narrowed between groups and why this is considered an improvement? I am not sure if figure 2C shows this.

Line 208-209. Please explain how these microbes were determined, by, for example, which test?

Line 223. It is not clear why the authors wanted to test SCFA and BA. This may be improved by further explaining the relationship between these two metabolite groups and the HH.

Line 240. Why do authors think that it is due to the small effect and sample size? It looks like there were more than 20 data points in those experiments.

Line 255. Please cite "Aitchison distance," if applicable.

Why were neurotransmitters tested? How about those data in NN groups.

Line 311-314. Please provide this information in the related result section, as it is essential to evaluate the rest of the microbiome data. Please also explain the reason for not being able to detect the probiotics in the microbiota data is. Is that because of the caveats with sequencing, sample collection, or library preparation, or do you think the microbes were washed out from rats in 28 days?

Line 318-320. Would you please provide a citation for this statement?

Line 335. Would you please provide a citation for this statement

Dear editor and reviewers,

We are submitting the revised manuscript “Gut microbiome-targeted modulations regulate metabolic profiles and alleviate altitude-related cardiac hypertrophy in rats” by Hu et al. [Paper #Spectrum01053-21].

We would like to thank the editor and the reviewers for their precious time and valuable comments. We have addressed each comment carefully and the responses are list below in blue. We believe that the manuscript has been substantially improved. Please let us know if you still have any question or concern about the manuscript.

Reviewer 1

The manuscript "Gut microbiome-targeted modulations regulate metabolic profiles and alleviate altitude-related cardiac hypertrophy in rats" by the authors Hu, and Pan et al. describes the effect of probiotics, prebiotics, and symbiotics in hypobaric hypoxia severity using a rat model. Due to the lack of research in 16S rRNA amplicon sequencing and fecal and plasma metabolome combination, the manuscript presents exciting findings, including prebiotics increasing neurotransmitter levels, which might be further investigated in another study. The manuscript is written very nicely and is easy to follow. I have some comments below.

Major Comments

1. Interestingly, HH increased three saturated medium-long chain fatty acids (palmitic, erucic, and behenic) levels, and symbiotics reduced their levels. Could the authors please explain the role of these fatty acids in hypobaric hypoxia development and microbiota dysbiosis?

To our knowledge, the levels of plasma free fatty acids are associated with the heart diseases (Khaw et al., 2012; Yamagishi et al., 2013). Previous studies consistently reported an increase in plasma palmitic acid (PA) under hypoxia environment (Cui et al., 2018; Liao et al., 2016; Sherwani et al., 2013). Elevated plasma PA was associated the development of cardiovascular diseases (Fatima et al., 2019). PA caused a decrease in insulin secretion and viability of islet cell in ex vivo (Sherwani et al., 2013). Plasma erucic acid (EA) was able to cross the blood-brain barrier and was incorporated into specific lipid metabolism via chain shortening (Golovko and Murphy, 2006). The roles of plasma EA and BA still remain poorly understood. Dietary EA could cause fatty accumulation in the heart (Abdellatif and Vles, 1970; Mattson and Streck, 1974). Dietary behenic acid (BA) suppressed triacylglycerol absorption and

prevented obesity in rats (Kojima et al., 2010; Moreira et al., 2017). But the dietary FFAs may not correlate to their changes or effects in plasma.

Symbiotics treatment reduced the levels of plasma PA, EA, and BA towards that in normal condition in HH rats, indicating the role of the gut microbiome in modulating the circulating FFAs. Correlation analysis revealed that these three fatty acids were positively associated with *Lactococcus garvieae*, *Bacteroides uniformis*, *Ruminococcus*, *Coprococcus*, etc; and negatively related to *Lactobacillus*, *Lactobacillus helveticus*, *Roseburia*, etc (Fig. 7). But it is not clear yet whether the alterations of the plasma FFAs are direct or indirect impact of gut microbiota.

2. The authors state that they could not identify the probiotics in the microbiota data after 28 days of treatment but also stated that the changes in amino acid levels in plasma are associated with the microbiota (lines 273-275). How do authors explain this?

We didn't detect the probiotics in feces based on the 16S amplicon sequencing data. The 16S data may not be sensitive enough to reflect the strain/species-level microbial change. We therefore utilized strain/species/genus-specific qPCR primers (Table below) and performed qPCR experiment to verify the probiotic bacteria. The sequences of some probiotics strains are not disclosed, here we detected two strains (*L.casei* Zhang and *L. plantarum P-8*) as representatives, as well as *L. rhamnosus*, *L. helveticus*, *B. adolescentis* species, *B. animalis* subsp. *lactis*, and *Bifidobacterium* genus.

Table. Primers of specific probiotics for quantitative-PCR analysis

No.	Targets	Specificity	Direction Sequence (Forward)	Direction Sequence (Reverse)	References
1	L. casei Zhang	Strain	CCGACGTACCAGCTCACT	TGAGCCGCTATCTGATAGTCTT	[PMID: 30409169]
2	L. plantarum P-8	Strain	ACTAACGGGAGGAGTGAT	ATAGTTCTCAAATCGGGAC	[PMID: 29867805]
3	L. rhamnosus	Species	GCCGATCGTTGACGTTAGTTGG	CAGCGTTATGCGATGCGAAT	[PMID: 32295530]
4	L. helveticus	Species	CTACTTCGCAGGCGTTAACT	GTA CTGTGATGCTCGCATACC	[PMID: 32295530]
5	B. animalis subsp. lactis	Species	GTGGAGACACGGTTTCCC	CACACCACACAATCCAATAC	[PMID: 14503690]
6	B. adolescentis	Species	CTCCAGTTGGATGCATGTC	CGAAGGCTTGCTCCAGT	[PMID: 14503690]
7	Bifidobacterium	Genus	CTCCTGGAACGGGTGG	GGTGTCTTCCCGATATCTACA	[PMID: 14503690]
8	16S rRNA Variable Region 3	Bacteria	CCTACGGGAGGCAGCAG	GTATTACCGCGGTGCTGG	[PMID: 30273610]

Significant higher levels of *L.casei Zhang* (Fig. A), *L.plantarum P-8* (Fig. B), *L.rhamnosus* (Fig. C), *B.animals sub. lactis* (Fig. E), and *Bifidobacterium* (Fig. G) were observed in HH-probiotics rats on day 28 than that on day 0. We also compared the abundance of *L.casei Zhang* (Fig. H), *B.animals sub. lactis* (Fig. I) and *Bifidobacterium* (Fig. J) on day 28 in both saline and probiotics groups. Their abundances were increased in probiotics group under both NN and HH conditions. The results together imply the presence of probiotics in feces.

Mantel test showed that fecal microbiome change was significantly correlated with plasma amino acids (Fig. 6A). Probiotics treatment also alleviated the shift of the plasma amino acids profile (Fig. 5A). These findings suggest that the changes in amino acid levels in plasma may be associated with probiotics-induced microbiome shift.

We have added the qPCR results to related result section.

3. The authors refer to multiple figures with one sentence, such as (Fig 1A-C). Please refer to each figure individually in the result section, such as Figure 1A and Figure 1B instead of Figure 1A-C.

We have referred each figure individually as the reviewer suggested. We hope it is now easier to read.

4. Please explain how were the prebiotics and probiotics used in this study were chosen?

The selected probiotics is a commercial product developed by professor Heping Zhang from the Inner Mongolia Agricultural University. This product composes of 3 *Bifidobacterium* strains and 6 *Lactobacillus* strains. The genome sequences and functions of the core strains have been substantially studied in humans and animals, such as *B. animalis subsp. lactis V9* (Peng et al., 2021; Sun et al., 2010; Yan et al., 2020), *L. casei Zhang* (He et al., 2020; Zhang et al., 2010; Zhu et al., 2021), *L. plantarum P-8* (Ma, 2018; Ma et al.,

2021; Wang et al., 2014; Zhang et al., 2015), and *L. helveticus H9* (Chen et al., 2014, 2015). We speculated that using the mixed strains maybe more conducive to identify whether intervention with probiotics is feasible to protect against high altitude heart disease.

Prebiotics we used in current study is also a commercially available product which is the nutritionally balanced mixture of galactose, inulin, and polydextrose. These prebiotics have been shown to improve diversity of gut microbiota, increase the abundance of dominant bacteria in the intestine which is extremely important for the health (Cheng et al., 2017).

Minor Comments

Line 30-31. Please clarify if the disruption of gut microbiota was ameliorated or partially ameliorated? Although the abstract reads very strong, the result section is written with more moderate language.

We apologize for the misleading expression. We used two expressions in the manuscript: “ameliorate” and “partially restore/reverse”. When we stated that “the treatment partially reversed the effect of hypobaric hypoxia on gut microbiota”, it means that the treatments ameliorated (not completely restored) the dysbiosis of gut microbiota. To ensure the expression were consistent and clear, we have modified “partially reverse/restore” in the manuscript to “ameliorate”.

Line 69-70. Please reconsider describing *Faecalibacterium*, *Lachnospiraceae* as health-promoting microbes as it reads like a vague statement without a solid scientific basis.

We have modified the sentence to “potential health-promoting microorganisms such as species belonging to *Faecalibacterium*, *Lachnospiraceae* [12, 13],

Erysipelotrichaceae [14] and *Ruminococcaceae* [15]”.

Line 76. I would avoid using aberrant microbiota as it is not scientifically classified.

We have modified the expression from “the aberrant gut microbiota” to “dysbiosis of gut microbiota”.

Line 77-78. Reference 21 (Bartolomaeus et al.) does not state any probiotic usage. They only used propionate. Please clarify if I am missing something.

We are very sorry for our negligence of losing a literature citation (Gómez-Guzmán et al., 2015). We have added it to the appropriate place.

Line 82. Please consider making the following change "gut microbiota is a"> "gut microbiota might be a."

We have made the change as suggested.

Line 89. What do you mean by had a greater impact? Compared to what other variables?

We are afraid that the reviewer may misread the word “great” as “greater”.

Line 92. Do you mean modulating dysbiosis in gut microbiota instead of "gut dysbiosis"? Or please explain what do you mean by "gut dysbiosis."

Yes, we have corrected the expression as suggested.

Figure 1B and C. Heat > Heart.

We apologize and have fixed this error.

Figure 1B and C. Could the author provide Individual data points for these figures as they did in Figures 1D and E?

We have made the change as suggested.

Line 720-722. Why do authors use two different symbols for the same p-value, such as ## and ** for $p < 0.001$?

In pharmacological research, two different symbols are often used to discriminate statistical comparison with the control group or the model group. We used * to represent significant difference between NN-saline group and other groups; and # to represent significant difference between HH-saline group and other group. If different symbols may cause confusion, we agree to only use * for $p < 0.05$, ** for $p < 0.01$.

Line 122. Please consider removing "mainly."

We have made the change as suggested.

Line 722. Please provide details of what statistical tests were used to compare each experimental group in Figure 1.

The statistical significance was performed by GraphPad Prism 8.0 software using one-way analysis of variance followed by Student's t-test between groups. We have added it to the figure 1 legend.

Line 200. Would you please delete "statistically"?

We have deleted the word as suggested.

Line 202-207. It is not clear what authors mean by diversity distances narrowed between groups and why this is considered an improvement? I am not sure if figure 2C shows this.

We apologize for our lack of clarity. As shown in Fig. 2C, the distances between NN and HH samples significantly decreased in three treatment groups compared to that in saline group, indicating that the gut microbiome in HH environment shifted towards normal compositions in the three treatment groups.

Line 208-209. Please explain how these microbes were determined, by, for example, which test?

Considering the age effect, we used log-ratio transformation and linear model to distinguish the differential ASVs between NN and HH in saline group, followed by multiple hypothesis testing correction with Benjamini-Hochberg method.

In detail, we fitted a linear model for each ASV with the formula:

$\log - ratio \sim NN/HH\ environment + probiotics + prebiotics + synbiotics$

where

$\log - ratio =$

$\log_{10}(\text{the relative abundance at day 28} / \text{the relative abundance at day 0})$.

The false discovery rate (FDR) of *NN/HH environment* variable was used to determine whether the ASV was significantly affected by HH. We have revised the methods accordingly.

Line 223. It is not clear why the authors wanted to test SCFA and BA. This may be improved by further explaining the relationship between these two metabolite groups and the HH.

Short-chain fatty acids (SCFAs) were the end products of fermentation of dietary fibers by the anaerobic intestinal microbiota. Bile acids (BAs) are synthesized from cholesterol in the liver and further metabolized by the gut microbiota into secondary bile acids. These metabolites have a range of effects locally in the gut and at both splanchnic and peripheral tissues which appear to affect the host metabolic and cardiovascular or heart health (Chambers et al., 2018; Jia et al., 2019).

Besides, our parallel study submitted to another journal, which investigated the longitudinal changes of gut microbiota during prolonged hypobaric hypoxia challenge, showed that HH significantly altered the abundance of genera *Parabacteroides*, *Alistipes*, *Lachnospira*, and *Prevotella*. Species belonging to these genera were reported to be involved in metabolism of SCFAs (Parker et al., 2020; Precup and Vodnar, 2019; Vanegas et al., 2017) or BAs (Wang et al., 2019). Taken together, we detected the SCFAs and BAs, as well as gut microbiota, to explore their relationships and the role of these metabolites in HH-induced cardiac hypertrophy.

Line 240. Why do authors think that it is due to the small effect and sample size? It looks like there were more than 20 data points in those experiments.

Each group (e.g. NN-saline group) contains 10 samples. PERMANOVA analysis showed that there was no significant difference between NN-saline and HH-saline for either fecal SCFA or BA profile. However, the difference between NN and HH samples was decreased in prebiotics group compared to saline group for SCFA profile (Fig. 4C); similarly, the difference between NN

and HH samples was decreased in prebiotics and synbiotics groups compared to saline group (Fig. 4D). The results suggest that there might be a certain degree of difference between NN and HH samples, but the effect size of the difference may be small. 10 samples may be not enough for a significant difference.

Line 255. Please cite "Aitchison distance," if applicable.

We have cited the reference as suggested.

Why were neurotransmitters tested? How about those data in NN groups.

Theoretically, during exposure to acute or chronic hypoxia, cells in the bilaterally-paired carotid bodies release neurotransmitters in peripheral plasma that stimulate afferent nerve fibers. The central projections of those fibers in turn activate cardiorespiratory centers in the brainstem, leading to an increase in ventilation and sympathetic drive that helps restore blood PO_2 , and finally maintain homeostasis in the respiratory and cardiovascular systems (Leonard et al., 2018). In this process, neurotransmitters, such as amino acid neurotransmitters and monoamine neurotransmitters, play an important role in responses of parasympathetic cardiac vagal neurons to hypoxia (Dergacheva et al., 2014). Indeed, substantial studies have demonstrated that vagus nerve regulates cardiac functions (Capilupi et al., 2020). Intriguingly, it has been shown that neurotransmitters and neuroactive metabolites generated from gut microbiota can enter peripheral circulation, affecting the vagus nerve activity and cardiorespiratory centers in the brainstem, and thereby regulate the cardiac function (Cryan et al., 2019). Therefore, it is plausible to detect the plasma neurotransmitters.

For the neurotransmitters in NN groups, PERMANOVA analysis based on

Aitchison distance showed that there was no significant difference between any NN-treatment group and NN-saline group.

Line 311-314. Please provide this information in the related result section, as it is essential to evaluate the rest of the microbiome data. Please also explain the reason for not being able to detect the probiotics in the microbiota data is. Is that because of the caveats with sequencing, sample collection, or library preparation, or do you think the microbes were washed out from rats in 28 days?

We believe we have addressed this comment in the previous response above. Briefly, qPCR amplification verified the presence of probiotics in feces. The qPCR result showed that the abundances of the probiotics were relatively low in probiotics-treated rats on day 28. Most of the gavaged probiotics may have been washed out at the collection timepoint. The low probiotic bacteria load, together with the low taxonomic resolution, may lead to the lack of sensitivity of 16S amplicon sequencing in detecting the probiotics strains.

We have added the qPCR results to related result section as suggested.

Line 318-320. Would you please provide a citation for this statement?

We have cited references as suggested.

Line 335. Would you please provide a citation for this statement?

We have cited references as suggested.

Reviewer 2

The manuscript “Gut microbiome-targeted modulations regulate metabolic profiles and alleviate altitude-related cardiac hypertrophy in rats” written by Hu et al. described treatment with probiotics, prebiotics, and symbiotic alleviating cardiac hypertrophy in the rat model of prolonged hypobaric hypoxia challenge, meanwhile, treatments also ameliorating gut microbial dysbiosis, and metabolic disruptions of certain metabolites in gut and plasma induced by hypobaric hypoxia. The study is interesting and the design is prudent. However, I have several concerns for the study.

1. Why this mix probiotics (3 *Bifidobacterium* strains and 6 *Lactobacillus* strains) were selected? Whether the single strain probiotics is more suitable?

The mix probiotics is a commercial product developed by professor Heping Zhang from the Inner Mongolia Agricultural University. This product composes of 3 *Bifidobacterium* strains and 6 *Lactobacillus* strains. The genome sequences and functions of the core strains have been substantially studied in humans and animals, such as *B. animalis subsp. lactis V9* (Peng et al., 2021; Sun et al., 2010; Yan et al., 2020), *L. casei Zhang* (He et al., 2020; Zhang et al., 2010; Zhu et al., 2021), *L. plantarum P-8* (Ma, 2018; Ma et al., 2021; Wang et al., 2014; Zhang et al., 2015), and *L. helveticus H9* (Chen et al., 2014, 2015).

There is no report on which probiotic strain may work for HH-induced microbiome change yet, thus we preferred to choose the combined probiotics with high colony forming units, which may be more likely to yield potent impacts on the gut microbiome than single strain.

2. Fig. 1 showed the size of hearts in different treatment group, which indicated the obvious difference. But the mRNA expression of some related genes in groups were confused. For example, treatment of prebiotics induced higher

expression of Collagen I, and no statistic difference of BNP in treatment groups compared with control group. I wonder whether other mechanisms regulated the heart in addition to these genes.

The mechanisms of pathological cardiac hypertrophy are complex. Other signaling pathways are also involved in physiological hypertrophy, including angiotensin II and endothelin 1, catecholamines and mTOR (Nakamura and Sadoshima, 2018).

In this study, we selected 6 typical gene markers of cardiac hypertrophy - ANP, BNP, α MHC, β MHC, collagen I, and collagen III. Different types of biomarkers represent distinct underlying mechanisms in the development of cardiac hypertrophy (Adroque et al., 2005). ANP and BNP serve as endogenous vasodilators to regulate blood pressure and restore normal hemodynamics (Cameron and Ellmers, 2003). Hypoxia-induced increases of ANP and BNP reflect physiological responses to pressure or volume overload (Adnot et al., 1988; Nakanishi et al., 2001). α MHC is the predominant isoform in the adult rat heart which has higher ATPase activity and contractile velocity than β MHC, the predominant isoform in the fetal rat heart (Morkin, 2000). The altered expressions of two isoforms (α MHC and β MHC) reflect the change of cardiac contractile function. Increased deposition of collagen and a shift in collagen types proportion have been considered as the hallmarks of myocardial fibrosis (González et al., 2019).

In current study, probiotics significantly increased α MHC expression in HH rats (Fig. 1J). Probiotics didn't significantly alter the expressions of these biomarkers, but it tended to reduce the expression of ANP, BNP and their ratio (Fig. 1G-I). We think that probiotics and prebiotics may regulate the HH-induced cardiac hypertrophy in different aspects. Synbiotics significantly

rescued the expression of ANP (Fig. 1G), α MHC (Fig. 1J) and the ratio of α MHC to β MHC (Fig. 1K), suggesting that there may be a synergistic effect between probiotics and prebiotics.

3. Fig. 2B only showed the difference between NN and HH, but for treatment group, no difference could be observed. Whether only analysis of HH data would show the difference of treatment group? Because the authors observed the phenotypes of different treatment groups in HH.

Yes, the treatment groups were significantly different with saline group in HH. Actually, in Fig. 2B, sample were also clustered by treatments, just not as obvious as the effect of HH. To better visualize the microbiome composition difference, we used HH samples at day 28 to draw a PCoA plot (see figure below). Samples of the same treatment were more similar than samples from different treatments. PERMANOVA analysis showed a significant difference between every two treatment groups (see table below).

distance_metric	day	treatment_1	treatment_2	p_value	significance
unweighted_unifrac	28	Saline	Probiotics	0.001	**
unweighted_unifrac	28	Saline	Prebiotics	0.001	**
unweighted_unifrac	28	Saline	Synbiotics	0.001	**
unweighted_unifrac	28	Probiotics	Prebiotics	0.001	**
unweighted_unifrac	28	Probiotics	Synbiotics	0.001	**
unweighted_unifrac	28	Prebiotics	Synbiotics	0.012	*

4. Bifidobacterium strains and Lactobacillus strains were given to rat everyday, can the authors detect them in the feces and whether the abundance changed?

This question is similar with the 2nd question from the first reviewer, please refer to the response there.

5. Three individual metabolites were identified which showed differentially abundant between NN and HH saline rats. Are there any metabolites which had differentially abundant between treatment group and control group?

While we understand the reviewer's concern, we would like to point out that we were more interested in the differential fecal metabolites between NN and HH groups and if the treatments could recover these differential metabolites, rather than the differential metabolites between treatment group and control group.

Given the interest of the reviewer, we further compared the log-ratio of fecal metabolites between saline and each treatment group under normoxia environment using t test followed by followed by multiple hypothesis testing correction with Benjamini Hochberg method. None of the fecal SCFA or BA

significantly differed between saline and treatment group with $FDR < 0.1$ (the significant threshold used in the manuscript).

6. Line 154-156 “All three treatments with probiotics, prebiotics, and synbiotics notably lessened this HH-induced ANP upregulation, even though only synbiotics group is statistically significant.” If there is no statistical difference, the authors can't conclude all three treatments notably lessened ANP upregulation.

We have revised the sentence to “The expression of ANP was slightly down-regulated after treatment with probiotics, prebiotics or synbiotics, even though only synbiotics group significantly lessened the ANP level”.

7. The English of the manuscript needs revision by a native English speaker.

We thank the reviewer for pointing out our language problem. We have the manuscript comprehensively checked by native English speakers.

References mentioned above

Abdellatif, A.M.M., and Vles, R.O. (1970). Pathological Effects of Dietary Rapeseed Oil in Rats. *Nutrition and Metabolism* *12*, 285–295.

Adnot, S., Chabrier, P.E., Brun-Buisson, C., Viossat, I., and Braquet, P. (1988). Atrial natriuretic factor attenuates the pulmonary pressor response to hypoxia. *J Appl Physiol* (1985) *65*, 1975–1983.

Adroge, J.V., Sharma, S., Ngumbela, K., Essop, M.F., and Taegtmeier, H. (2005). Acclimatization to chronic hypobaric hypoxia is associated with a differential transcriptional profile between the right and left ventricle. *Mol Cell Biochem* *278*, 71–78.

Cameron, V.A., and Ellmers, L.J. (2003). Minireview: natriuretic peptides during development of the fetal heart and circulation. *Endocrinology* *144*, 2191–2194.

Capilupi, M.J., Kerath, S.M., and Becker, L.B. (2020). Vagus Nerve Stimulation and the Cardiovascular System. *Cold Spring Harb Perspect Med* *10*, a034173.

Chambers, E.S., Preston, T., Frost, G., and Morrison, D.J. (2018). Role of Gut Microbiota-Generated Short-Chain Fatty Acids in Metabolic and Cardiovascular Health. *Curr Nutr Rep* *7*, 198–206.

Chen, Y., Liu, W., Xue, J., Yang, J., Chen, X., Shao, Y., Kwok, L., Bilige, M., Mang, L., and Zhang, H. (2014). Angiotensin-converting enzyme inhibitory activity of *Lactobacillus helveticus* strains from traditional fermented dairy foods and antihypertensive effect of fermented milk of strain H9. *J Dairy Sci* *97*, 6680–6692.

Chen, Y., Zhang, W., Sun, Z., Meng, B., and Zhang, H. (2015). Complete genome sequence of *Lactobacillus helveticus* H9, a probiotic strain originated from kurut. *J Biotechnol* *194*, 37–38.

Cheng, W., Lu, J., Li, B., Lin, W., Zhang, Z., Wei, X., Sun, C., Chi, M., Bi, W., Yang, B., et al. (2017). Effect of Functional Oligosaccharides and Ordinary Dietary Fiber on Intestinal Microbiota Diversity. *Front Microbiol* *8*, 1750.

Cryan, J.F., O’Riordan, K.J., Cowan, C.S.M., Sandhu, K.V., Bastiaanssen, T.F.S., Boehme, M., Codagnone, M.G., Cusotto, S., Fulling, C., Golubeva, A.V., et al. (2019). The Microbiota-Gut-Brain Axis. *Physiological Reviews* *99*, 1877–2013.

Cui, X., Ye, L., Li, J., Jin, L., Wang, W., Li, S., Bao, M., Wu, S., Li, L., Geng, B., et al. (2018). Metagenomic and metabolomic analyses unveil dysbiosis of gut microbiota in chronic heart failure patients. *Sci Rep* *8*, 635.

Dergacheva, O., Weigand, L.A., Dyavanapalli, J., Mares, J., Wang, X., and Mendelowitz, D. (2014). Function and modulation of premotor brainstem parasympathetic cardiac neurons that control heart rate by hypoxia-, sleep-, and sleep-related diseases including obstructive sleep apnea. *Progress in Brain Research* *212*.

Fatima, S., Hu, X., Gong, R.-H., Huang, C., Chen, M., Wong, H.L.X., Bian, Z., and Kwan, H.Y. (2019). Palmitic acid is an intracellular signaling molecule involved in disease development. *Cell. Mol. Life Sci.* *76*, 2547–2557.

Golovko, M.Y., and Murphy, E.J. (2006). Uptake and metabolism of plasma-derived erucic

acid by rat brain. *Journal of Lipid Research* 47, 1289–1297.

Gómez-Guzmán, M., Toral, M., Romero, M., Jiménez, R., Galindo, P., Sánchez, M., Zarzuelo, M.J., Olivares, M., Gálvez, J., and Duarte, J. (2015). Antihypertensive effects of probiotics *Lactobacillus* strains in spontaneously hypertensive rats. *Mol Nutr Food Res* 59, 2326–2336.

González, A., López, B., Ravassa, S., San José, G., and Díez, J. (2019). The complex dynamics of myocardial interstitial fibrosis in heart failure. Focus on collagen cross-linking. *Biochimica et Biophysica Acta (BBA) - Molecular Cell Research* 1866, 1421–1432.

He, Q., Hou, Q., Wang, Y., Shen, L., Sun, Z., Zhang, H., Liang, M.-T., and Kwok, L.-Y. (2020). Long-term administration of *Lactobacillus casei* Zhang stabilized gut microbiota of adults and reduced gut microbiota age index of older adults. *Journal of Functional Foods* 64, 103682.

Jia, Q., Li, H., Zhou, H., Zhang, X., Zhang, A., Xie, Y., Li, Y., Lv, S., and Zhang, J. (2019). Role and Effective Therapeutic Target of Gut Microbiota in Heart Failure. *Cardiovascular Therapeutics* 2019, e5164298.

Khaw, K.-T., Friesen, M.D., Riboli, E., Luben, R., and Wareham, N. (2012). Plasma Phospholipid Fatty Acid Concentration and Incident Coronary Heart Disease in Men and Women: The EPIC-Norfolk Prospective Study. *PLOS Medicine* 9, e1001255.

Kojima, M., Tachibana, N., Yamahira, T., Seino, S., Izumisawa, A., Sagi, N., Arishima, T., Kohno, M., Takamatsu, K., Hirotsuka, M., et al. (2010). Structured triacylglycerol

containing behenic and oleic acids suppresses triacylglycerol absorption and prevents obesity in rats. *Lipids in Health and Disease* *9*, 77.

Leonard, E.M., Salman, S., and Nurse, C.A. (2018). Sensory Processing and Integration at the Carotid Body Tripartite Synapse: Neurotransmitter Functions and Effects of Chronic Hypoxia. *Front Physiol* *9*, 225.

Liao, W.-T., Liu, B., Chen, J., Cui, J.-H., Gao, Y.-X., Liu, F.-Y., Xu, G., Sun, B.-D., Zhang, E.-L., Yuan, Z.-B., et al. (2016). Metabolite Modulation in Human Plasma in the Early Phase of Acclimatization to Hypobaric Hypoxia. *Sci Rep* *6*, 22589.

Ma, C. (2018). The regulation mechanism of *Lactobacillus plantarum* P-8 on intestinal microflora. *Chin. Sci. Bull.* *64*, 298–306.

Ma, T., Jin, H., Kwok, L.-Y., Sun, Z., Liong, M.-T., and Zhang, H. (2021). Probiotic consumption relieved human stress and anxiety symptoms possibly via modulating the neuroactive potential of the gut microbiota. *Neurobiol Stress* *14*, 100294.

Mattson, F.H., and Streck, J.A. (1974). Effect of the Consumption of Glycerides Containing Behenic Acid on the Lipid Content of the Heart of Weanling Rats. *The Journal of Nutrition* *104*, 483–488.

Moreira, D.K.T., Santos, P.S., Gambero, A., and Macedo, G.A. (2017). Evaluation of structured lipids with behenic acid in the prevention of obesity. *Food Research International* *95*, 52–58.

Morkin, E. (2000). Control of cardiac myosin heavy chain gene expression. *Microsc Res Tech* *50*, 522–531.

Nakamura, M., and Sadoshima, J. (2018). Mechanisms of physiological and pathological cardiac hypertrophy. *Nat Rev Cardiol* *15*, 387–407.

Nakanishi, K., Tajima, F., Itoh, H., Nakata, Y., Osada, H., Hama, N., Nakagawa, O., Nakao, K., Kawai, T., Takishima, K., et al. (2001). Changes in atrial natriuretic peptide and brain natriuretic peptide associated with hypobaric hypoxia-induced pulmonary hypertension in rats. *Virchows Arch* *439*, 808–817.

Parker, B.J., Wearsch, P.A., Veloo, A.C.M., and Rodriguez-Palacios, A. (2020). The Genus *Alistipes*: Gut Bacteria With Emerging Implications to Inflammation, Cancer, and Mental Health. *Frontiers in Immunology* *11*, 906.

Peng, C., Yao, G., Sun, Y., Guo, S., Wang, J., Mu, X., Sun, Z., and Zhang, H. (2021). Comparative effects of the single and binary probiotics of *Lactocaseibacillus casei* Zhang and *Bifidobacterium lactis* V9 on the growth and metabolomic profiles in yogurts. *Food Research International* 110603.

Precup, G., and Vodnar, D.-C. (2019). Gut *Prevotella* as a possible biomarker of diet and its eubiotic versus dysbiotic roles: a comprehensive literature review. *British Journal of Nutrition* *122*, 131–140.

Sherwani, S.I., Aldana, C., Usmani, S., Adin, C., Kotha, S., Khan, M., Eubank, T., Scherer, P.E., Parinandi, N., and Magalang, U.J. (2013). Intermittent Hypoxia Exacerbates

Pancreatic β -Cell Dysfunction in A Mouse Model of Diabetes Mellitus. *Sleep* *36*, 1849–1858.

Sun, Z., Chen, X., Wang, J., Gao, P., Zhou, Z., Ren, Y., Sun, T., Wang, L., Meng, H., Chen, W., et al. (2010). Complete Genome Sequence of Probiotic *Bifidobacterium animalis* subsp. *lactis* Strain V9. *Journal of Bacteriology* *192*, 4080–4081.

Vanegas, S.M., Meydani, M., Barnett, J.B., Goldin, B., Kane, A., Rasmussen, H., Brown, C., Vangay, P., Knights, D., Jonnalagadda, S., et al. (2017). Substituting whole grains for refined grains in a 6-wk randomized trial has a modest effect on gut microbiota and immune and inflammatory markers of healthy adults. *Am J Clin Nutr* *105*, 635–650.

Wang, K., Liao, M., Zhou, N., Bao, L., Ma, K., Zheng, Z., Wang, Y., Liu, C., Wang, W., Wang, J., et al. (2019). *Parabacteroides distasonis* Alleviates Obesity and Metabolic Dysfunctions via Production of Succinate and Secondary Bile Acids. *Cell Rep* *26*, 222-235.e5.

Wang, L., Zhang, J., Guo, Z., Kwok, L., Ma, C., Zhang, W., Lv, Q., Huang, W., and Zhang, H. (2014). Effect of oral consumption of probiotic *Lactobacillus planatarum* P-8 on fecal microbiota, SIgA, SCFAs, and TBAs of adults of different ages. *Nutrition* *30*, 776-783.e1.

Yamagishi, K., Folsom, A.R., Steffen, L.M., and ARIC Study Investigators (2013). Plasma fatty acid composition and incident ischemic stroke in middle-aged adults: the Atherosclerosis Risk in Communities (ARIC) Study. *Cerebrovasc Dis* *36*, 38–46.

Yan, Y., Liu, C., Zhao, S., Wang, X., Wang, J., Zhang, H., Wang, Y., and Zhao, G. (2020).

Probiotic *Bifidobacterium lactis* V9 attenuates hepatic steatosis and inflammation in rats with non-alcoholic fatty liver disease. *AMB Express* *10*, 101.

Zhang, W., Yu, D., Sun, Z., Wu, R., Chen, X., Chen, W., Meng, H., Hu, S., and Zhang, H. (2010). Complete Genome Sequence of *Lactobacillus casei* Zhang, a New Probiotic Strain Isolated from Traditional Homemade Koumiss in Inner Mongolia, China. *J Bacteriol* *192*, 5268–5269.

Zhang, W., Sun, Z., Bilige, M., and Zhang, H. (2015). Complete genome sequence of probiotic *Lactobacillus plantarum* P-8 with antibacterial activity. *J Biotechnol* *193*, 41–42.

Zhu, H., Cao, C., Wu, Z., Zhang, H., Sun, Z., Wang, M., Xu, H., Zhao, Z., Wang, Y., Pei, G., et al. (2021). The probiotic *L. casei* Zhang slows the progression of acute and chronic kidney disease. *Cell Metab* *33*, 1926-1942.e8.

November 30, 2021

Dr. Zhenjiang Zech Xu
Nanchang University
State Key Laboratory of Food Science and Technology
Nanchang, Jiangxi
China

Re: Spectrum01053-21R1 (Gut microbiome-targeted modulations regulate metabolic profiles and alleviate altitude-related cardiac hypertrophy in rats)

Dear Dr. Zhenjiang Zech Xu:

Your manuscript has been accepted, and I am forwarding it to the ASM Journals Department for publication. You will be notified when your proofs are ready to be viewed.

Sincerely,

Jennifer Auchtung
Editor, Microbiology Spectrum

Journals Department
supplemental material: Accept